# Hectometric scale simulations of a Mediterranean heavy precipitation event during HyMeX SOP1

Olivier Nuissier[1], Fanny Duffourg[2], Maxime Martinet[1], Véronique Ducrocq[1], and Christine Lac[1]

[1]CNRM (Météo-France & CNRS), 42 avenue G. Coriolis 31057 Toulouse Cédex
[2]CSG (CNES & Agence spatiale européenne)

**Correspondence:** Olivier Nuissier (olivier.nuissier@meteo.fr)

**Abstract.** Offshore convection occurred over the Mediterranean sea on 26 October 2012 and was well documented during the first Special Observation Period (SOP1) of the Hydrological cycle in the Mediterranean Experiment (HyMeX). This paper analyses the triggering and organizing factors involved in this convection case study, and examines how they are simulated and represented at hectometric resolutions. For that purpose, a Large Eddy Simulation (LES) of this real case study is carried out with a 150 m horizontal resolution over a large domain encompassing the convective systems, as well as the low level flow feeding convection over the sea. This LES is then compared to a reference simulation performed with a 450 m grid spacing in the heart of the so-called "grey zone" of turbulence modelling.

Increase of horizontal resolution from 450 m down to 150 m is unable to reduce significantly, for this case study, deficiencies of the simulation, more related to an issue of initial and lateral boundary conditions. Indeed, some of the triggering factors, such as a converging low level flow driven by a surface low pressure system, are simulated quite similarly for both simulations. However, differences for other mechanisms still exist since greater surface precipitation amounts are simulated at 450 m. It is found that the entrainment process, characterized by small eddies at the cloud edges, is strongly underestimated at 450 m horizontal resolution, missing the mixing with the environmental air. Therefore a too rapid development of deep convection is simulated at this horizontal resolution, associated with fast-track microphysical processes and enhanced dynamics. Whereas at 150 m horizontal resolution, the updraft cores are mainly resolved, as well as the subsiding shell, while subgrid eddies, produced by dynamical processes, are localized at the cloud interior edges better representing the entrainment process.

Furthermore, this first LES of a real Mediterranean precipitating case study highlights a convective organisation with very fine scale features within the converging low level flow, features that are definitively out of range of models with kilometric horizontal resolutions.

# 1 Introduction

Regularly during fall, Heavy Precipitation Events (HPEs) occur over the Northwestern Mediterranean basin and more particularly over the mountainous coastal regions of France, Spain and Italy. In most cases, large amounts of precipitation are recorded in less than one day (typically more than 200 mm in less than 24 h, and sometimes in only few hours) when a mesoscale convective system (MCS) develops and stays over the same area for several hours (Nuissier et al., 2008; Buzzi et al., 2014; Davolio et al., 2016; Duffourg et al., 2018, among others).

Several past case studies, especially in the framework of the HyMeX Special Observation Period (SOP1), have investigated extensively the mechanisms and physical processes associated with such high-impact events, based on numerical experiments with kilometric horizontal resolutions and dedicated observations (Ducrocq et al., 2016). Over the Western Mediterranean region, HPEs evolve often within a favourable synoptic pattern, including an upper level trough centred over the Iberian Peninsula and/or high values of geopotential anchored over Central Europe, driving a persistent low-level moist and conditionally unstable marine flow directed towards the coastal mountainous regions. Due to regional and local effects, the low level flow is then deflected by neighbouring mountain ranges (Pyrénées, Massif Central and Alps) or islands such Corsica and Sardinia. In order to trigger deep convection, a lifting mechanism is needed to focus and finally release the conditional convective instability at the same location. Orographic lifting of the conditionally unstable low level marine flow impinging the foothills neighbouring the Western Mediterranean is a well-studied mechanism for renewing convection triggering at the same location (Barthlott and Davolio, 2016; Duffourg et al., 2018). Lifting can also be due to local convergence in the low-level circulation induced by the orography of the region (Barthlott et al., 2016; Buzzi et al., 2014; Scheffknecht et al., 2016) or lee cyclogeneses (Jansa et al., 2001; Duffourg et al., 2016). A low-level cold pool, possibly forming under the MCS, can also lift the low-level flow at its leading edge (Ducrocq et al., 2008) and/or modify the low-level circulation locally and enhance convergence areas (Duffourg et al., 2016).

While these previous have shown that a horizontal resolution of about one kilometer is able to simulate many of the observed features of Mediterranean HPEs, as well as their associated key physical mechanisms, they have difficulties representing the time at which convection is triggered and its organization. The increase of horizontal resolution to sub-kilometric grid spacings could be a way for improvement. Indeed, Verrelle et al. (2015) have shown, for an idealized convective case, that an increase of horizontal resolution of 500 m simulates stronger updraughts within convective cells associated with a greater cloudy coverage. Hanley et al. (2015) and Fiori et al. (2017) carried out numerical modeling HPE case studies over Italy with grid spacings ranging from 5 km to 200 m. Similar to idealized studies, they found that smaller convective cells with a stronger intensity are simulated with a horizontal resolution finer than 500 m. Scheffknecht et al. (2016) found that, when the horizontal grid spacing is reduced to 500 m, precipitation is more widespread, maximum values are lower, and individual convective cells are smaller. These previous studies concluded that surface precipitation amounts are enhanced and convective structures are better represented at sub-kilometric scale.

Nevertheless, the increase in horizontal resolution may pose problems especially for the turbulence parameterization within the so-called "grey zone" (GZ), typically in the range of horizontal resolution of about a few hundreds of meters and also

depending on the model's effective resolution (Wyngaard, 2004). Indeed, there are still uncertainties about how turbulence should be modeled at these resolutions. Past studies have investigated the impact of a one-dimensional (1D) versus a three-dimensional (3D) representation of the turbulence at sub-kilometric horizontal resolution. Verrelle et al. (2015) showed that a 3D turbulence leads to stronger mixing and greater cloud cover. Machado and Chaboureau (2015) carried out simulations using 1D turbulence which produced too many small cloud systems and rainy cells with a shorter lifespan. Additionally, Verrelle et al. (2017) compared kilometric deep convection simulations, up to 500 m resolution, to LES and showed that subgrid turbulent kinetic energy at these resolutions was underestimated with the eddy-diffusivity turbulence scheme, due to an underestimation of thermal turbulence production while resolved vertical velocities tend to be overestimated. Based on simulations carried out with 500 m grid spacings, Martinet et al. (2017) found that, for a specific case study of HyMeX SOP1 (IOP16a), not only cloud organisation but also the simulated environment and processes governing convection are strongly sensitive to the formulation of the mixing length. Indeed, when turbulent mixing is weak (i.e. weak subgrid turbulent kinetic energy), the resolved winds are increased, leading to greater low level moisture advection, higher hydrometeor contents, marked low level cold pools, and therefore more intense simulated convective systems.

This strong sensitivity obtained by Martinet et al. (2017) motivates the adoption of a Large Eddy Simulation (LES)-like approach, enabling a more suitable representation of turbulence within and at the edge of convective clouds and also within the atmospheric boundary layer. In a LES framework, eddies that contain most of the kinetic energy are resolved whereas smaller eddies, that carry less than 20% of the total kinetic energy, are represented by subgrid processes. Although several studies have demonstrated the need to use grid spacing of about 100 m to represent the convective flow correctly (Bryan et al., 2003; Petch, 2006; Stein et al., 2015; Zängl et al., 2015; Dauhut et al., 2015, 2016, among others), most of these works were carried out using an idealized framework and/or using a small domain. To the authors's knowledge no LES numerical simulation of a real case study of Mediterranean HPE over a large domain has previously been performed. The purpose of this paper is to evaluate and analyse the impact of increasing horizontal resolution to LES in a numerical simulation of Mediterranean HPE, using a true topography in addition to realistic initial and forcing conditions. The present study goes further analysing how the physical mechanisms and convective organization are represented from sub-kilometric horizontal resolutions down to LES. For that purpose, a large domain encompassing the convective systems as well as the low level flow feeding convection over the sea is considered. The paper focuses on the same offshore convection case study described in Duffourg et al. (2016) and Martinet et al. (2017) (IOP16a), which took place on 26 October 2012 during the SOP1 of the HyMeX field programme (Ducrocq et al., 2014).

This article is organized as follows. A quick review of the case study and the involved mechanisms are provided in section 2. The numerical model and the simulation setup are presented in section 3. Both simulations with hectometric horizontal resolutions and LES, respectively, are compared in terms of rainfall field analysis and convection organization in section 4. The physical processes at cloud scale leading to very deep convection and convection organization are assessed in section 5. Finally, the study is summarized and conclusions are given in section 6, where perspectives for future work are also suggested.

## 2 The IOP16a case study

### 2.1 Meteorological conditions

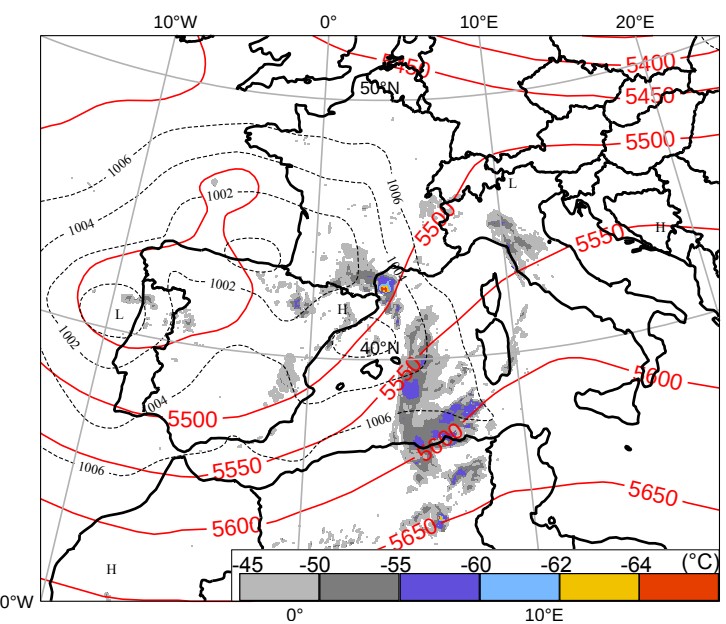

**Figure 1.** ARPEGE analysis for IOP16a in terms of 500 hPa geopotential height (solid line, every 50 dam) and mean sea level pressure (dashed line, hPa) and Meteosat Second Generation infrared brightness temperature (°C) (channel 9, 10.8 $\mu$m) (coloured areas) valid at 0600 UTC on 26 October 2012.

This section presents the HPE observed on 26 October 2012 and well documented during the HyMeX SOP1. During this event, a large part of the Northwestern Mediterranean was concerned by intense precipitation which locally led to flash flooding.

Figure 1 shows the synoptic scale situation, valid at 0600 UTC on 26 October 2012. The synoptic situation was characterized by a large deep upper-level low centred over Spain. A short-wave trough along with an associated potential vorticity anomaly (not shown) circulated ahead of the main system (i.e. offshore intense convective systems), passing over southeastern Spain, France and then Italy. Moreover, a large low is also anchored over Spain as seen with the mean sea level pressure (Figure 1).

At low levels, Figure 2 shows that deep convection was embedded within a very moist flow all along the system life-cycle.

A surface low pressure formed and strengthened downstream of the Iberian mountainous regions, between Spain and Balearic Islands. It was strongly associated with the eastward propagation of the upper level trough.

Such favourable meteorological conditions described above led to the generation of several convective systems over the Mediterranean Sea. The first convective cells appeared just East of the Spanish coast around 0600 UTC on 26 October. Convection started to organize while moving northeastwards and forming an intense south-north oriented line (marked CS1) (Figure

2b). It appeared that, in addition to offshore convection, the meteorological situation was also favourable for orographic forcing

over the southern slopes of the Massif Central throughout the day on 26 October 2012. The southermost cells behind CS1 also developed and organized into a second mesoscale convective system (marked CS2), which headed eastnortheastwards towards the southeastern coastal regions of France. In the afternoon of 26 October 2012, CS2 crossed the region with surface rainfall amounts of about 150 mm in 24h causing two casualties.

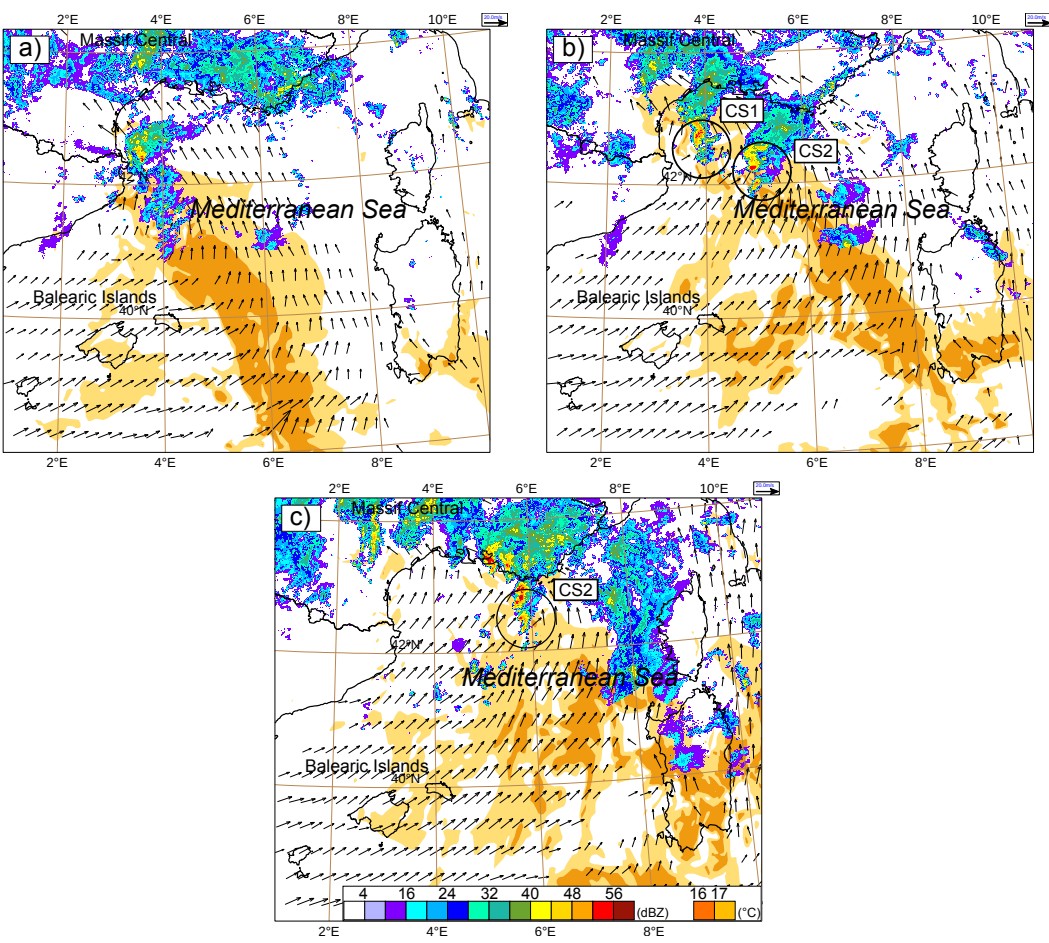

**Figure 2.** Observed radar reflectivity (colour scale, dBZ) superimposed to AROME analysis for IOP16a in terms of adiabatic wet-bulb potential temperature (coloured areas) and horizontal winds greater than 20 m s$^{-1}$ (arrows) at 925 hPa valid at, **(a)** 0600 UTC, **(b)** 0900 UTC and **(c)** 1200 UTC on 26 October 2012.

## 2.2    Triggering mechanisms

An exhaustive evaluation of Meso-NH simulations for the convective systems involved in IOP16a can be found either in Duffourg et al. (2016) or Martinet et al. (2017) at 2.5 km and 500 m horizontal resolution, respectively.

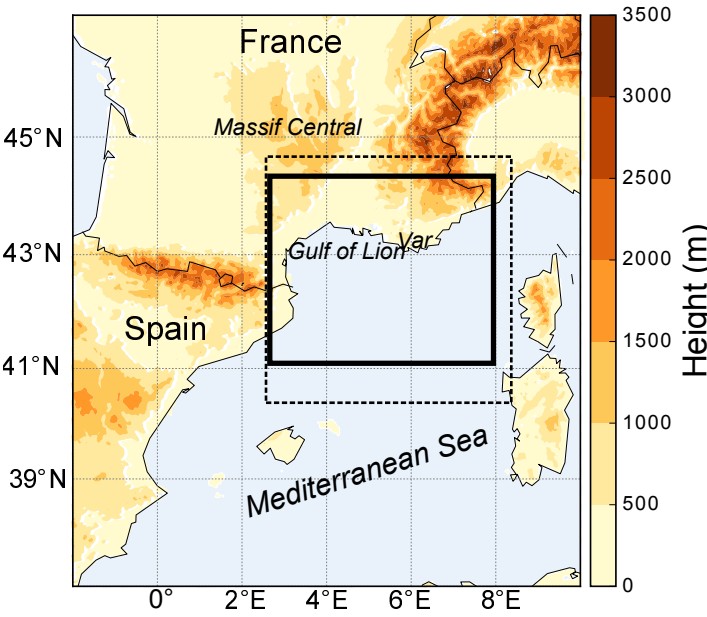

**Figure 3.** Computational domains used for the outer/coarser domain (900 m, dashed rectangle) and for the inner/finer domain (450 m or 150 m, thick solid rectangle), respectively (see text for more explanation). The model orography (m) is shown in shading.

The numerous dedicated airborne and ground-based observations during the HyMeX SOP1 (suites of water vapour lidars, wind profilers, radiosoundings and boundary-layer drifting balloons, among others) over the sea and along the coast of the northwestern Mediterranean offered a unique framework for validating the convective systems simulated over the sea by kilometric scale numerical models initialized and driven by kilometric resolution analyses. Indeed, Duffourg et al. (2016) showed
5    that these convective systems during IOP16a were fed during their evolution over the sea by moist and conditionnally unstable air masses. A southwest to southeasterly converging low level flow over the sea is the main triggering mechanism acting to continually initiate and maintain the renewal of convective cells, contributing to the back-building systems CS1 and CS2. Lifting is also partly due to evaporative low level cooling. In addition it appears that this low level cooling also controls the organization into a mesoscale convective system.
10    Martinet et al. (2017) found that the mechanisms mentioned above, as well as the dynamics of the convective systems, are sensitive to the mixing length formulation used in turbulence parameterization, at horizontal resolution of 500 m. These elements motivate an increase of the horizontal resolution up to LES scale in this present study to assess how the involved physical mechanisms are represented

## 3 Description on the numerical experiments

### 3.1 The Meso-NH model

The French non-hydrostatic mesoscale numerical model Meso-NH (Lac et al., 2018) was used for the simulation of the IOP16a case study. The Gal-Chen and Somerville (1975) vertical coordinate is used with 140 vertical levels. The vertical grid spacing is stretched with altitude, from 10 m close to the surface to 250 m aloft. The top of the domain is at 20 km altitude, and a Rayleigh damping is progressively applied above 15 km altitude to the perturbations of the wind components and the thermodynamical variables with respect to their large-scale values, in order to prevent spurious reflections from the upper boundary.

The prognostic variables are the three Cartesian components of velocity, the dry potential temperature, the different water mixing ratios and the turbulent kinetic energy. Pressure perturbations are determined by solving the elliptic equation obtained by combining air mass continuity and momentum conservation equations. The transport scheme for momentum variables is the Weighted Essentially NonOscillatory (WENO) scheme (Shu and Osher, 1988) of the $5^{th}$ order combined with the fourth order Runge-Kutta time-splitting method (Lunet et al., 2017), while the other 20 variables are transported with the Piecewise Parabolic Method (PPM) scheme (Colella and Woodward, 1984). A bulk one-moment mixed microphysical scheme (Caniaux et al., 1994; Pinty and Jabouille, 1998) governs the equations of the six water species: water vapour, cloud water, rain water, primary ice, snow aggregates, and graupel. The turbulence parametrization is based on a 1.5-order closure (Cuxart et al., 2000) and the calculation of the turbulent flow is performed through a three dimensional (3D) scheme for horizontal resolution below kilometric grid spacings. For horizontal resolution of about a few hundred meters and coarser grids, the mixing length follows the method of Bougeault and Lacarrère (1989) whereas for LES resolution, the mixing length formulation follows the one proposed by Deardorff (1972), which is directly proportional to the grid volume. The numerical set-up used in this study also used other parametrization schemes including the Rapid Radiation Transfer Model parametrization (Mlawer et al., 1997), the Pergaud et al. (2009) Eddy Diffusivity Mass Flux scheme for shallow convection and the surface model SURFace EXternalisé (SURFEX) (Masson et al., 2013).

### 3.2 Simulation design

In this present study a LES of the IOP16a case study is carried out with the Meso-NH model. As in Duffourg et al. (2016) or Martinet et al. (2017), the initial and lateral boundary conditions of the simulations are provided by the AROME-WMED analyses (Fourrié et al., 2015). Since it is not suitable to initialize and drive the LES simulation using directly the AROME-WMED analyses due to a too large horizontal resolution gap, an intermediate domain with a coarser grid is used, through the nesting technique in a two-way interactive mode (Stein et al., 2000). The coarser grid provides the lateral boundary conditions to the finer one, while the variables of the coarser grid are relaxed with a short relaxation time toward the finer grid's values in the overlapping area. Vertical grids are the same. For this present study, two nested domains in the horizontal plane with horizontal grid spacings of 900 m and 150 m (hereafter **HR150**) were used. Another control simulation (hereafter **LR450**) is carried out with horizontal grid spacings of 900 m and 450 m, respectively (Figure 3). The simulation domains cover nearly

the same regions as in Martinet et al. (2017), i.e. the southeastern France and the northwestern Mediterranean, encompassing the precipitating systems and their marine low level moisture supplying flow (see Figure 3).

The simulations are initialized at 0000 UTC on 26 October 2012 and run until 1200 UTC. At 0000 UTC and for a 3h- period only the coarser domain with a grid spacing of 900 m (dashed rectangle in Figure 3) is started . The finer domains (450 m or
150 m) are activated at 0300 UTC and for the rest of the simulation period.

## 4   Overview of the numerical simulations

This section aims at assessing the impact of increasing horizontal resolution by comparing both simulations over the same spatial scale. For that purpose the reference LES fields are upscaled at a coarser horizontal resolution of 450 m (hereafter **LR150**). The fields are thus coarse grained by averaging the LES fields as in Verrelle et al. (2017) or Honnert et al. (2011). We focus
on the life-cycle of heavy precipitation occurring over the sea mainly in the morning of 26 October 2012, with a mature phase happening between 0600 UTC and 1200 UTC. During this period, the triggering mechanisms within the convective systems are examined.

### 4.1   Comparaison between LR450 and LR150 simulations

#### 4.1.1   Precipitation field analysis

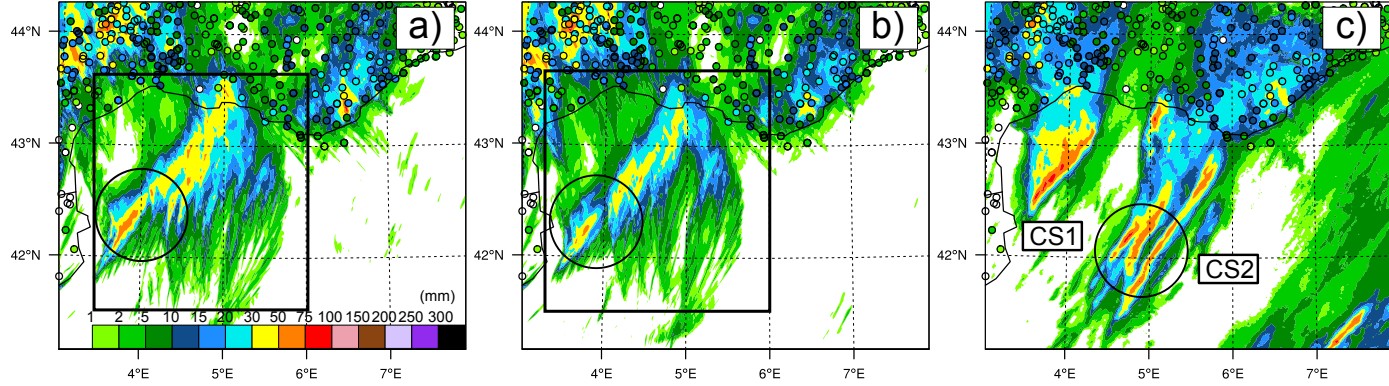

**Figure 4.** 6-h accumulated simulated and observed surface precipitation from, **(a) LR450** experiment, **(b) LR150** experiment, and **(c)** radar estimates, respectively between 0600 UTC and 1200 UTC on October 2012. The small circles on panels represent the raingauge observations whereas the largest ones highlight the surface precipitation associated with CS2.

Figure 4 represents the surface rainfall accumulated between 0600 UTC and 1200 UTC on 26 October 2012. Both **LR450** and **LR150** simulate fairly well the areas of strong precipitation over Southeastern France. The surface rainfall over land is

well reproduced in terms of magnitude and location over the Var region and over the southeastern part of the Massif Central, compared to observations (Figure 4).

As for precipitation over the sea, it is worth mentioning that radar quantitative precipitation estimation is impacted by large uncertainties, and in any case should be viewed cautiously. Nevertheless, two areas of strong accumulated surface rainfall are observed over the sea in Figure 4c; a first one located near 4°E and a second one a few tens of kilometers southwest near 5°E. These regions of large precipitation are caused by the convective systems CS1 and CS2 mentioned earlier. Although only one convective system is simulated (i.e. CS2), its evolution over the sea is well simulated by both **LR450** and **LR150**, except for a location too far west and stronger offshore rainfall just east of Spanish coast (see area circled in Figure 4). This precipitation pattern is explained by a former convective system triggered earlier in the simulation and not dissipating in time. But, the spatial distribution and magnitude of precipitation appear stronger for **LR450** compared to **LR150**. The results presented here for **LR450** are fairly comparable to those obtained by Martinet et al. (2017) with a 500 m horizontal resolution. Differences are probably due to both different initial conditions and numerical schemes in both simulations.

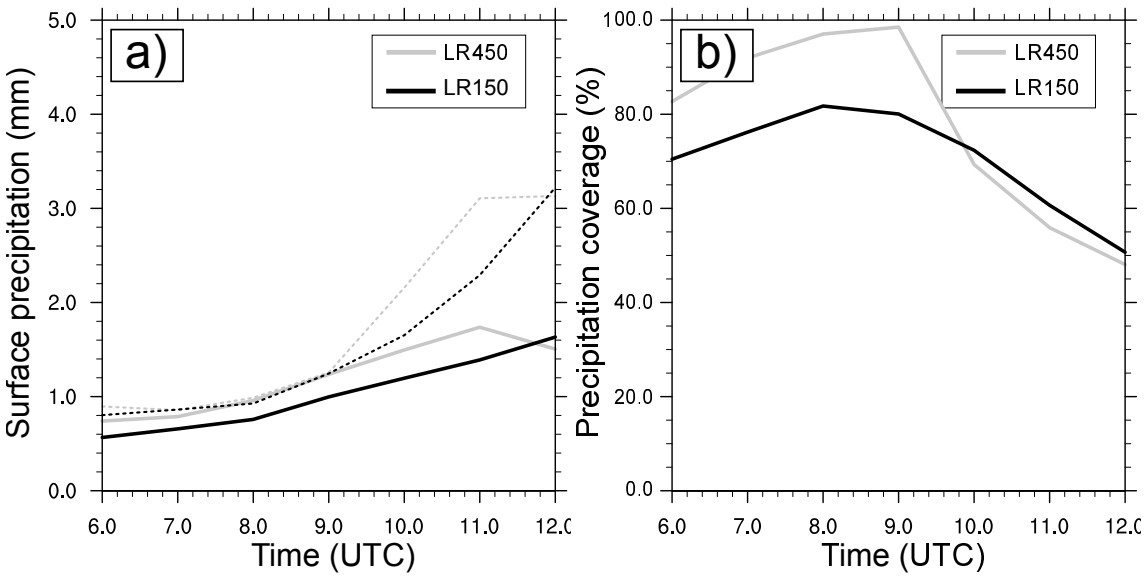

**Figure 5.** (a) Time series of the hourly surface precipitation (mm) for **LR450** and **LR150** experiment, averaged over the whole subdomain seen in Figure 4 (solid lines) and averaged over the area where surface precipitation exceeds 0.1 mm h$^{-1}$ (dashed lines). (b) Time series of spatial coverage (%) of the hourly surface precipitation exceeding the threshold of 0.1 mm h$^{-1}$ over the same subdomain.

In order to assess the consistency of these results along all the simulation period, time series of surface precipitation, averaged over the subdomain represented in Figure 4, are calculated. This domain encompasses the evolution of the convective system over the sea and not taking account here precipitation on land. Figure 5 shows the surface precipitation averaged over the whole subdomain, as well as averaged over the area where surface precipitation exceeds 0.1 mm h$^{-1}$.

One can remark that the surface precipitation simulated by **LR450** is greater than **LR150** from 0600 until near 1100 UTC on 26 October 2012 (Figure 5a). However, it must be emphasized that the largest surface rainfall in **LR450** from 0600 UTC until 0900 UTC are mainly due to more spatially widespread precipitation, whereas between 0900 UTC and 1100 UTC stronger rainfall rates contribute more to the largest precipitation for **LR450**. The sensitivity of these results to precipitation threshold has also been examined (not shown). Indeed, higher precipitation thresholds ($> 20$ mm h$^{-1}$) confirm that **LR450** simulates stronger rainfall over a greater spatial area after 0900 UTC.

These results with more simulated precipitation with a horizontal resolution of 450 m, compared to the LES, are somewhat different than those obtained by Fiori et al. (2017) or Hanley et al. (2015) for instance, who found that the largest surface rainfall amounts are simulated at the finest scale. Nevertheless around 1200 UTC, the surface rainfall simulated by **LR150** becomes greater (Figure 5a).

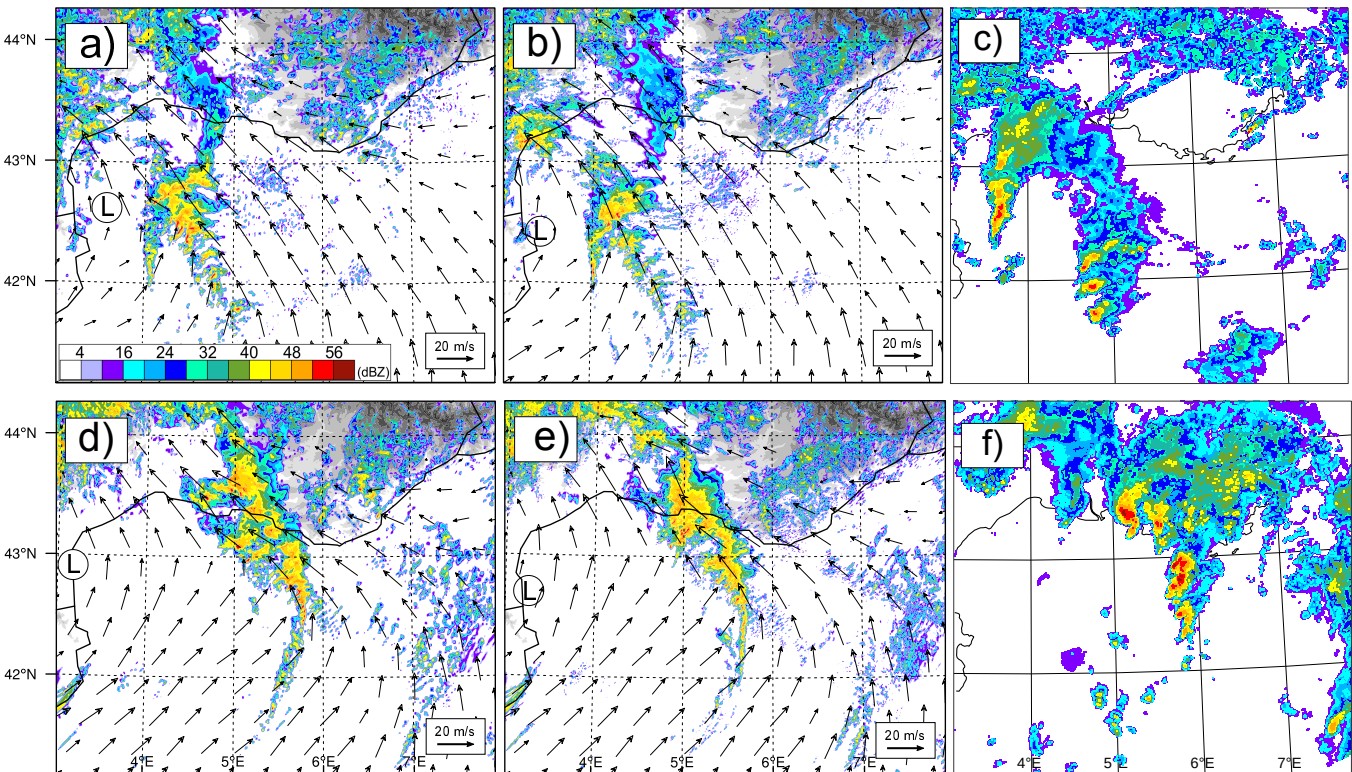

**Figure 6.** Simulated radar reflectivities (dBZ) at 850 hPa and horizontal winds (m s$^{-1}$) at 950 hPa from : **(a) LR450** and **(b) LR150** at 0900 UTC on 26 October 2012, respectively. Panels **(d)** and **(e)** are the same except for 1200 UTC. The symbol L marks the location of the surface low pressure system. Panels **(c)** and **(f)** represent the observed radar reflectivities at 0900 UTC and 1200 UTC, respectively.

Both **LR450** and **LR150** are now compared by analysing the time evolution of the convection over the sea. It is worth mentioning that a convective system is triggered earlier shortly after the beginning of the simulation and was maintained in the simulation for too long. This former convective system is responsible for the large surface precipitation accumulation just east of the Spanish coasts (Figure 4). The first convective cells of interest are actually triggered a bit late in both simulations near 0700 UCT on 26 October 2012 (not shown), leading to a different behaviour and evolution during the triggering stage compared to observations.

Figure 6 represents the simulated radar reflectivities at 0900 UTC and 1200 UTC at 850 hPa compared to observations, respectively, as well as horizontal winds at 950 hPa. As mentioned previously, the circulation, in which the convective system evolves, was characterized by strong low level convergence, controlled by a surface low pressure located between Spain and Balearic Islands. This pattern enhances locally convergence in the southwesterly to southeasterly low level flow. This surface low pressure is simulated quite similarly in both **LR450** and **LR150** (Figure 6b and d). The mature stage is characterized by continual renewal of convective cells along the low level convergence around 0900 UTC as shown in Figure 6a and b. This particular organization is also visible in the observed radar reflectivities (Figure 6c). It appears that the convective system develops faster in **LR450** with convection extending more northeastwards. As a matter of fact, when comparing both simulations at upper levels, i.e. analysing the infrared brightness temperature, one can see a more spatially extended convective system with a more pronounced cloudy anvil in **LR450** (Figure 7a and b). In the other hand, one can also remark that, for both simulations **LR450** and **LR150**, the coldest area in terms of brightness temperature is less spatially extended compared to the observations (Figure 7c and f). This is probably related to a lack of iced hydrometeors at upper levels in the simulations. As the surface low pressure moves east-northeastwards, the low level flow strengthens and convergence increases, organizing into a very pronounced line. Near 1200 UTC on 26 October 2012, differences between both **LR450** and **LR150** become barely discernable and precipitating structures are more comparable (Figure 6c and d). Both simulations show at this time a convective system tilting along a slight southwest-northeast axis that is fairly comparable with the observed radar reflectivities (Figure 6f).

In summary, increase of horizontal resolution from 450 m until 150 m for this case does not significantly improve some deficiencies of the simulation. Indeed, the low level convergence could be closely controlled by the surface low pressure, and both **LR450** and **LR150** handle its strengthening very similarly. Moreover, only a single convective system is simulated in both **LR450** and **LR150** instead of two compared to observations, and too strong surface rainfall is simulated just east of Spanish coast. Nevertheless, significant differences appear between both simulations during the mature stage of the convective systems, especially in terms of accumulation of surface precipitation, spatial extent of the simulated systems, intensity (between 0900 UTC and 1100 UTC) and development of convection over the Mediterranean Sea.

### 4.1.2 Triggering mechanisms

In this section, the differences highlighted in terms of simulated surface precipitation patterns, convective organization and intensity are explained by analysing the environment and the mechanisms associated with the convective systems. It has been shown in the previous section that the mature stage is simulated differently by both **LR450** and **LR150**. The system of interest is triggered around 0700 UTC just east of another decaying one, offshore of the Balearic Islands, slowly moving

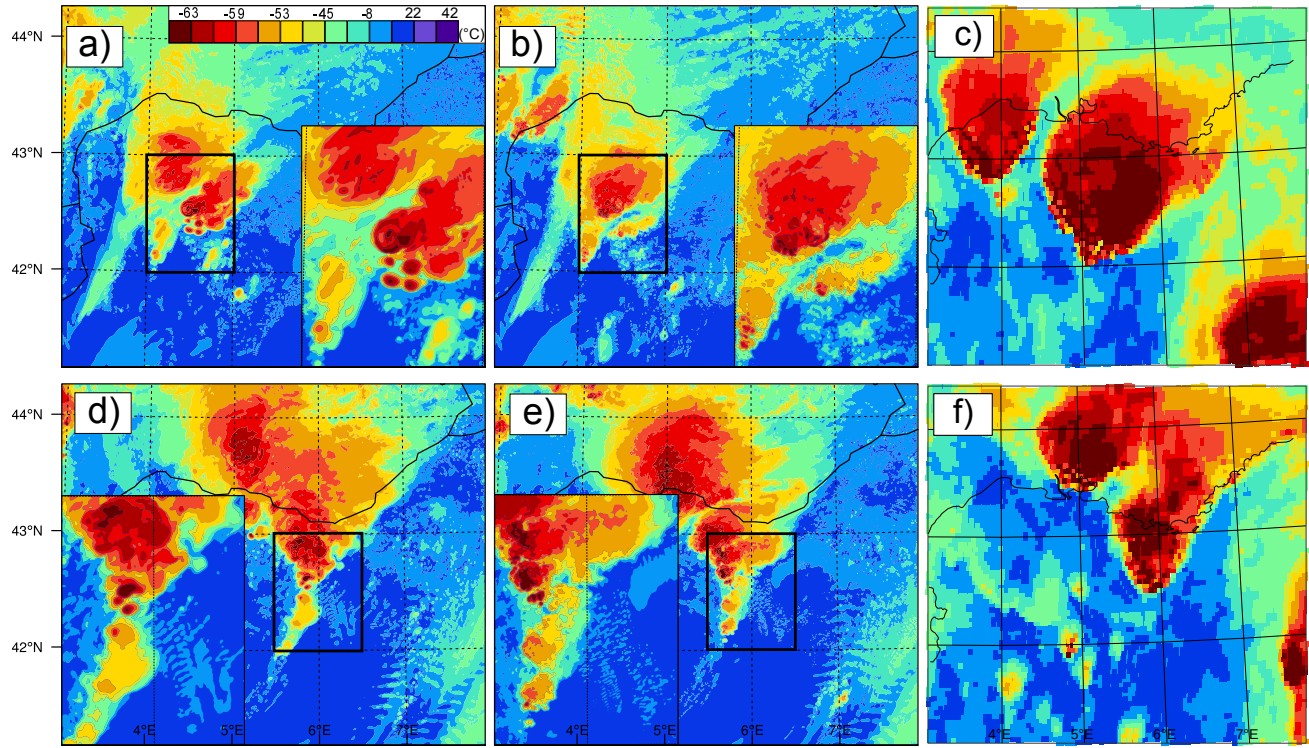

**Figure 7.** Same as Figure 6 except for simulated infrared brightness temperature (°C).

east-northeastwards over the sea. This convective system developed within a warm and moist environment and high values of conditional convective instability (not shown) along a low level convergence line, as discussed previously. These triggering mechanisms are quite similar to those found by Duffourg et al. (2016) and Martinet et al. (2017) with a 2.5 km or 500 m horizontal resolution, albeit with a slight time lag.

Thereafter and as also discussed previously, the systems simulated in both simulations adopt different behaviours during their mature stage. Figure 8 shows the low level moisture flux integrated in the first 3000 m of the troposphere, the virtual potential temperature at near 40 m AGL (third model level), as well as updraughts with magnitude greater than 3 m s$^{-1}$ at 500 m height, simulated for both **LR450** and **LR150** around the mature stage. The spatial distribution of the low level moist flux is simulated differently in both simulations. Indeed around 0900 UTC, **LR150** simulates stronger values downstream of the

convective system, whereas intense values of low level moisture flux are located in the vicinity of the precipitating event for **LR450**, with a maximum value reaching near 500 kg m$^{-2}$ s$^{-1}$ (Figure 8a and b). Near 1200 UTC on 26 October 2012, the low level moisture flux becomes stronger and more uniform for **LR150**, throughout the convective system (Figure 8d), with locally stronger fluxes than **LR450**. Strong and pronounced updraughts are simulated for both simulations at the leading edge of the area of low level convergence, thus evidencing the lifting of conditionally unstable air masses triggering deep convection. This

area of low level convergence and high moisture flux appears slightly northeast for **LR150**.

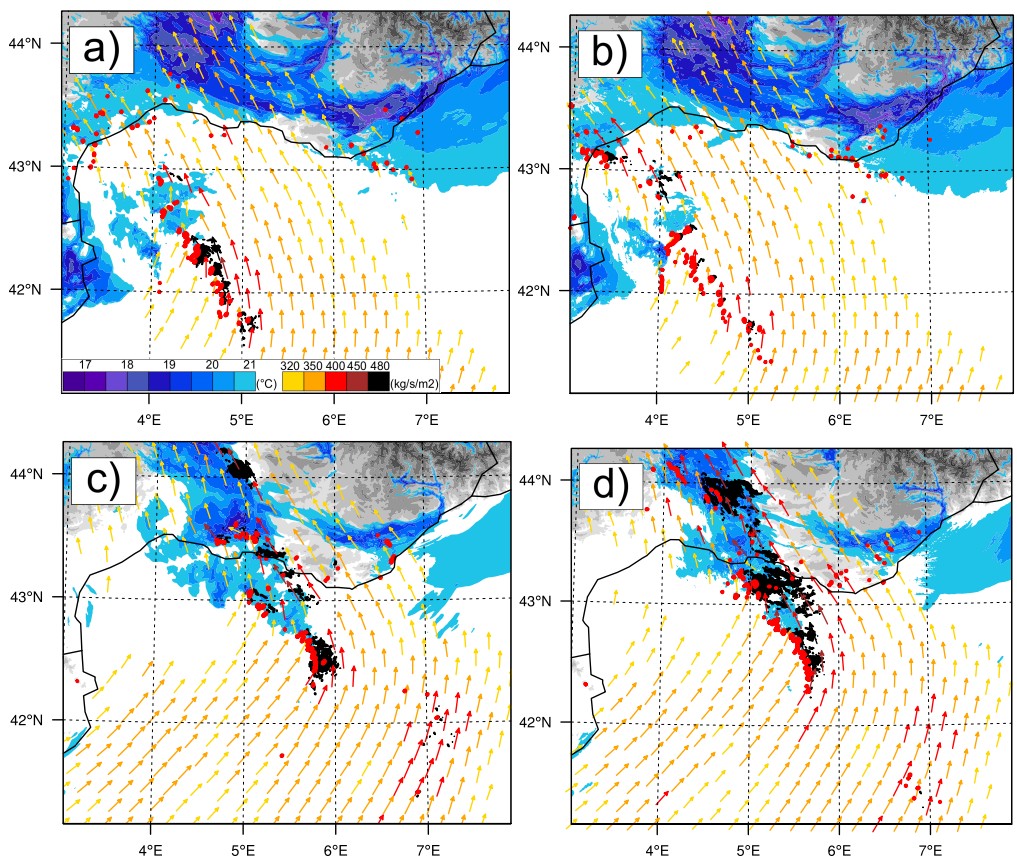

**Figure 8.** Same as Figure 6 except for virtual potential temperature at the third model level (near 40 m AGL) (°C, blue areas), water vapour horizontal flux integrated over the lowest 3000 m ASL (kg m$^{-2}$ s$^{-1}$, arrows) and vertical velocity exceeding 3 m s$^{-1}$ at 500 m ASL (red isolines). Black areas denote values of moisture flux exceeding 480 kg m$^{-2}$ s$^{-1}$.

Another mechanism responsible for lifting is also present and competes with the low level convergence. Indeed, both simulations reproduce a low level cold pool (LLCP) underneath the convective system. However, the cooling is more spatially widespread with stronger horizontal gradients greater than 2°C for **LR450**, favouring more lifting. The LLCP interacts with the low level flow and also enhances locally the area of convergence, as shown in Duffourg et al. (2016) with their simulations at 2.5 km horizontal resolution. The less intense LLCP for **LR150** probably leads to a less deflected flow, stronger moisture advection and triggering of convection downstream compared to the system simulated by **LR450**.

In order to assess how these differences on mechanisms impact the dynamics of the simulated convective systems, Figure 9 presents time series of the mixing ratios for the precipitating hydrometeors as well as the strongest updraughts, averaged over the whole subdomain shown in Figure 4. Precipitating hydrometeors (rain, graupel and snow aggregates) contents and updraughts are increasing throughout the morning of 26 October 2012 until 1200 UTC, showing a strengthening of the convective

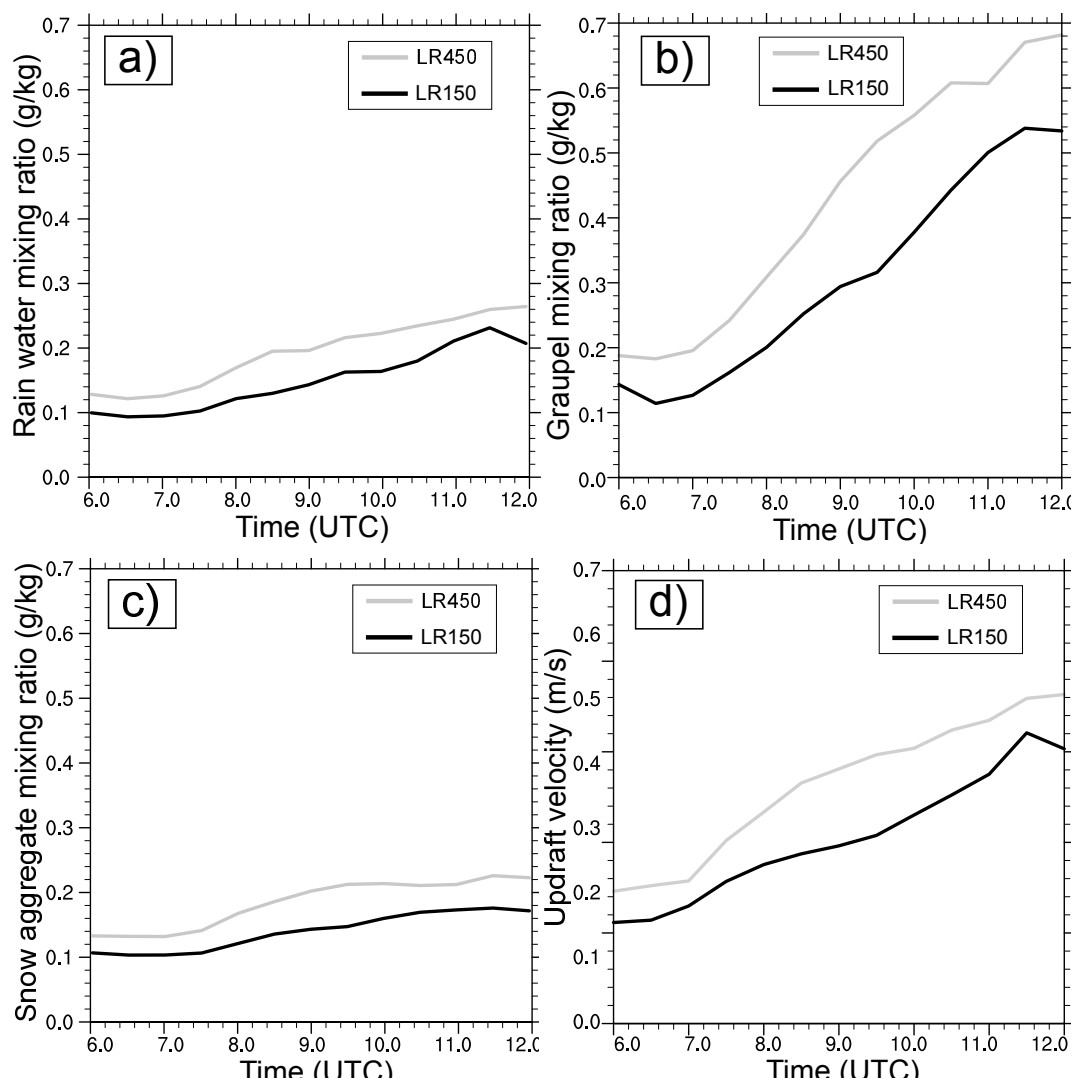

**Figure 9.** Time series of : (**a**) rain water mixing ratio, (**b**) graupel mixing ratio, (**c**) snow aggregate mixing ratio, and (**d**) the $90^{th}$ percentile of vertical velocities, for **LR450** and **LR150** experiments respectively, averaged over the whole subdomain seen in Figure 4.

system with intense dynamics. Hydrometeors contents and vertical motions are systematically greater for **LR450**, exceeding sometimes by nearly 50 % the **LR150** values (Figure 9b).

Dynamics of the convective system and the hydrometeor contents, leading to the LLCP and more surface precipitation, appear thus much greater in the simulation with a horizontal resolution of 450 m compared to the LES. However, when horizontal winds and water vapour contents are examined, the low level environment for both simulations appear quite similar, even with a slightly weaker low level flow for **LR450** (not shown). Therefore, one can argue that, for this specific IOP16a

case study, the largest surface precipitation obtained for the simulation with a horizontal resolution of 450 m could be partly explained by the manner in which dynamics and microphysics are represented within the convective system and its interactions with the near surrounding air.

## 5   Very fine-scale convective organization

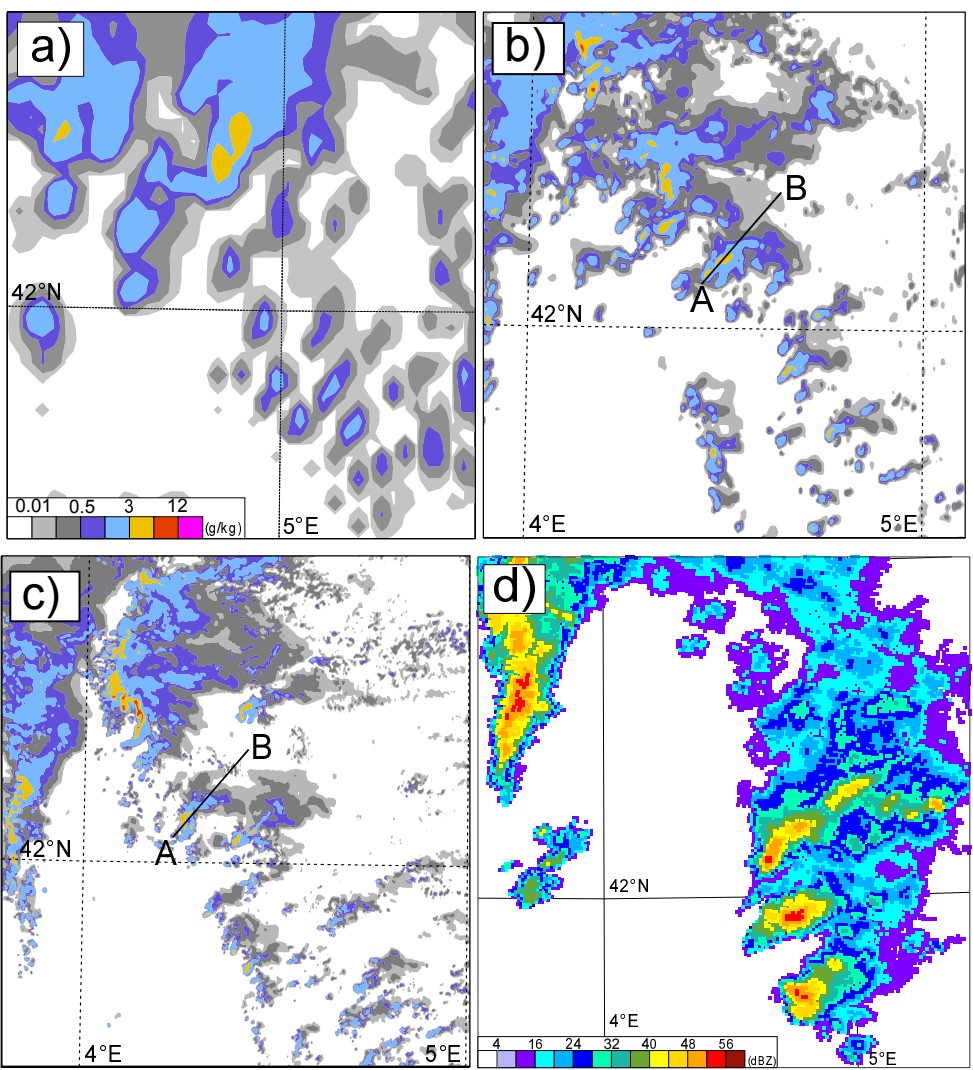

**Figure 10.** Zoomed view of the spatial distribution of warm microphysical species (sum of rain and cloud water mixing ratio) at 1500 m height for : **(a)** Duffourg et al. (2016)'s simulation at 2.5 km horizontal resolution, **(b) LR450**, and **(c) HR150**, respectively. Panel **(d)** shows the observed radar reflectivities at 0900 UTC on 26 October 2012.

In previous section 4, both simulations were compared by upscaling the LES at the same resolution as **LR450**. It has been shown that the increase in horizontal resolution does not modify significantly the environment of the precipitating system. The triggering mechanisms, such as the low level convergence, are not modified significantly, except for the LLCP which is strengthened by possible enhancement of rainfall evaporation at low-levels in the convective system.

5    In the LES, the precipitating structures are simulated and represented at a finer scale. It offers the possibility of analysing the dynamics more precisely and the convective organization down to the cell scale. Therefore, in the rest of the paper, the LES simulation will be presented at the native horizontal resolution (**HR150**). Figure 10 highlights the spatial distribution of the rain and cloud mixing ratios at 1500 m height for both **LR450** and **HR150**, which are compared to the Duffourg et al. (2016)'s simulation at 2.5 km horizontal resolution. If the kilometric simulation develops a convective structure within the low level 10   convergence, it is unable to organize small convective cells, as depicted at hectometric resolution (Figure 10a compared to Figure 10b,c). This simulated organisation with trains of very small convective cells are also depicted and confirmed by radar observations (Figure 10d).

   Within the southermost part of the precipitating system, one can see intense convective cell trains, oriented southwest to northeast, triggered and organised along the low level convergence. Figure 11 shows a vertical cross section along a convective 15   cell between 0745 UTC and 0815 UTC (A-B axis in Figure 10), in both **LR450** and **HR150**, of the simulated precipitating hydrometeor contents (rain, graupel and snow aggregate contents) and the non precipitating (cloud and ice) water contents. Vertical motions (updraughts and downdraughts) are also represented. At the beginning of the developing stage at 0745 UTC, the non precipitating hydrometeors depict for **LR450** a rather shallow cloud with a base below 500 m ASL (Above Sea Level). There are some mixed hydrometeor contents appearing at 0800 UTC but they are limited to 5-6 km height (Figure 11c). Very 20   quickly - 15 min later - at 0815 UTC, strong upward motions are simulated within the cell, associated with large graupel contents up to 9 km height (Figure 11e). When reaching the top of troposphere, a cloudy anvil forms and fans out downstream of the upper level southwesterly flow.

   However, the cloud appearance is quite different and especially more realistic for **HR150**. Indeed, the structure is still organized with pronounced hydrometeor amounts advected upwards within convective updraughts but, in this case, there 25   are stronger gradients of vertical velocity located all along the edge of the convective cell, i.e. downward motions in the environment neighbouring and abutting strong updraughts at the edge cloud. Moreover in **HR150**, the cloud takes a clearly discernible cumuliform appearance throughout all the period of the developing stage, that might be an indication of a better representation of cloud-edge entrainment and therefore a better representation, at this scale, of horizontal turbulent mixing between the cloud and its environment (Figure 11b,d and f).

30    In order to illustrate that point, Figure 12, 13 and 14 show horizontal cross sections of vertical velocities, subgrid turbulent kinetic energy (TKE) at 6 km and 8 km height, and both dynamical (DP) and thermal (TP) contributions in the TKE production at 6 km height, throughout the convective cells displayed in Figure 11, respectively. As one can see in Figure 12a and 12c and Figure 13a and 13c , at 450 m horizontal resolution, the updraft cores are partly resolved, as vertical velocities do not exceed 12 m s$^{-1}$, and are also partly unresolved, with TKE sometimes higher than 30 m$^2$ s$^{-2}$. This TKE is produced by both dynamical 35   and thermal processes, and is mainly localized within the updraft cores. The eddies near the cloud edges, which are subgrid at

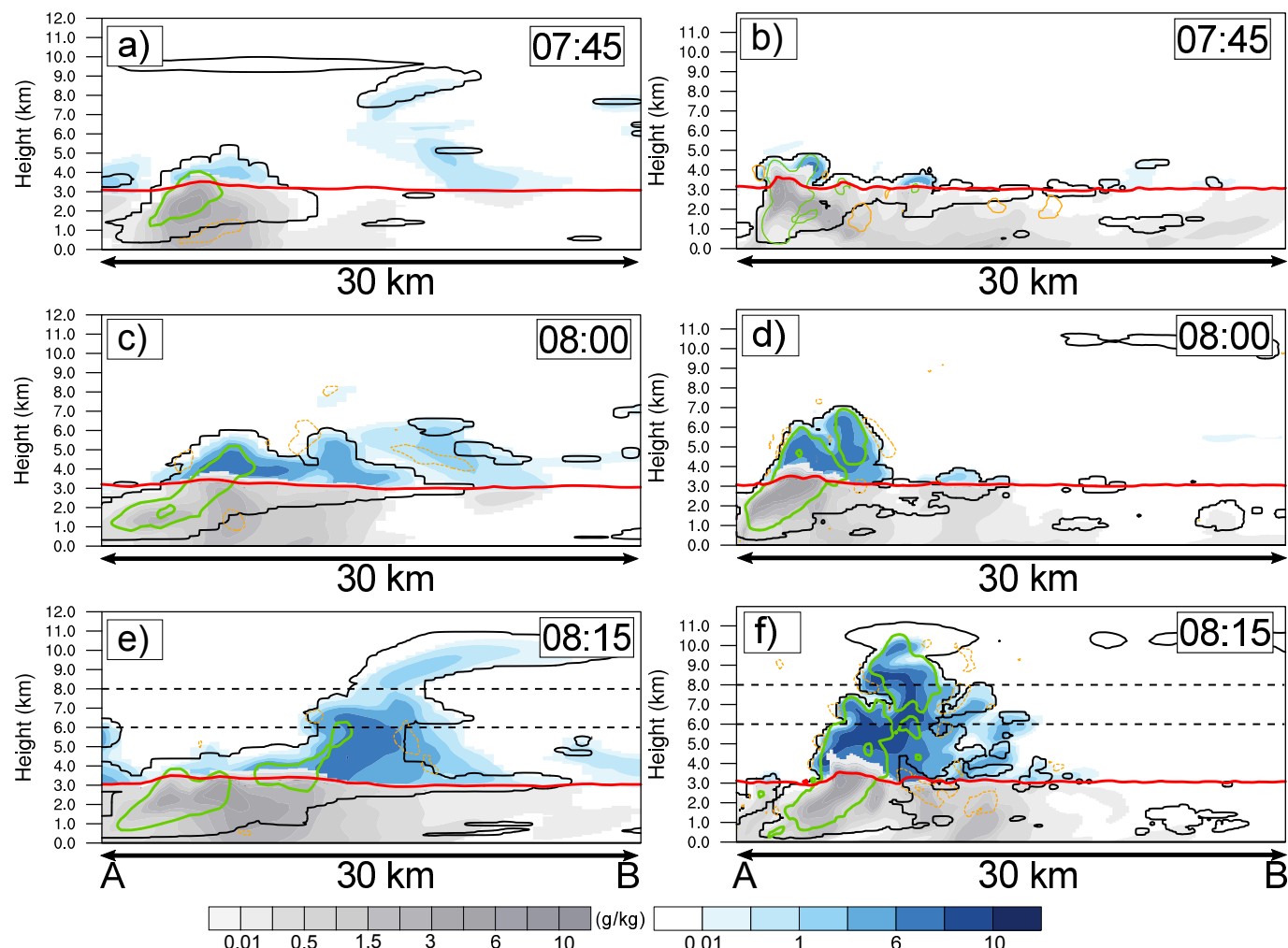

**Figure 11.** Vertical cross section along a convective cell for : **(a) LR450**, and **(b) HR150** at 0745, 0800 and 0815 UTC, respectively. Rain water contents (g kg$^{-1}$) are represented in grey, whereas mixed precipitating hydrometeor (graupel and snow aggregate) contents (g kg$^{-1}$) are highlighted in blue. The black solid line delineates the cloud boundaries (threshold of cloud and ice water contents > 0.001 g kg$^{-1}$). Updraughts (green lines) and downdraughts (orange dashed lines) are represented with a threshold of about 5 m s$^{-1}$ and -2 m s$^{-1}$, respectively. The red solid line stands for the freezing level.

450 m horizontal resolution, are not represented by the turbulence scheme. These results are in a good agreement with those obtained by Verrelle et al. (2017) and Strauss et al. (2019), who have shown that a commonly used eddy-diffusivity turbulence scheme underestimates the TKE at kilometric and hectometric (500m) horizontal resolution, especially at the cloud edges but also in the updraft cores where the thermal production is misrepresented as the scheme does not enable the countergradient structures present in the updraft to be reproduced.

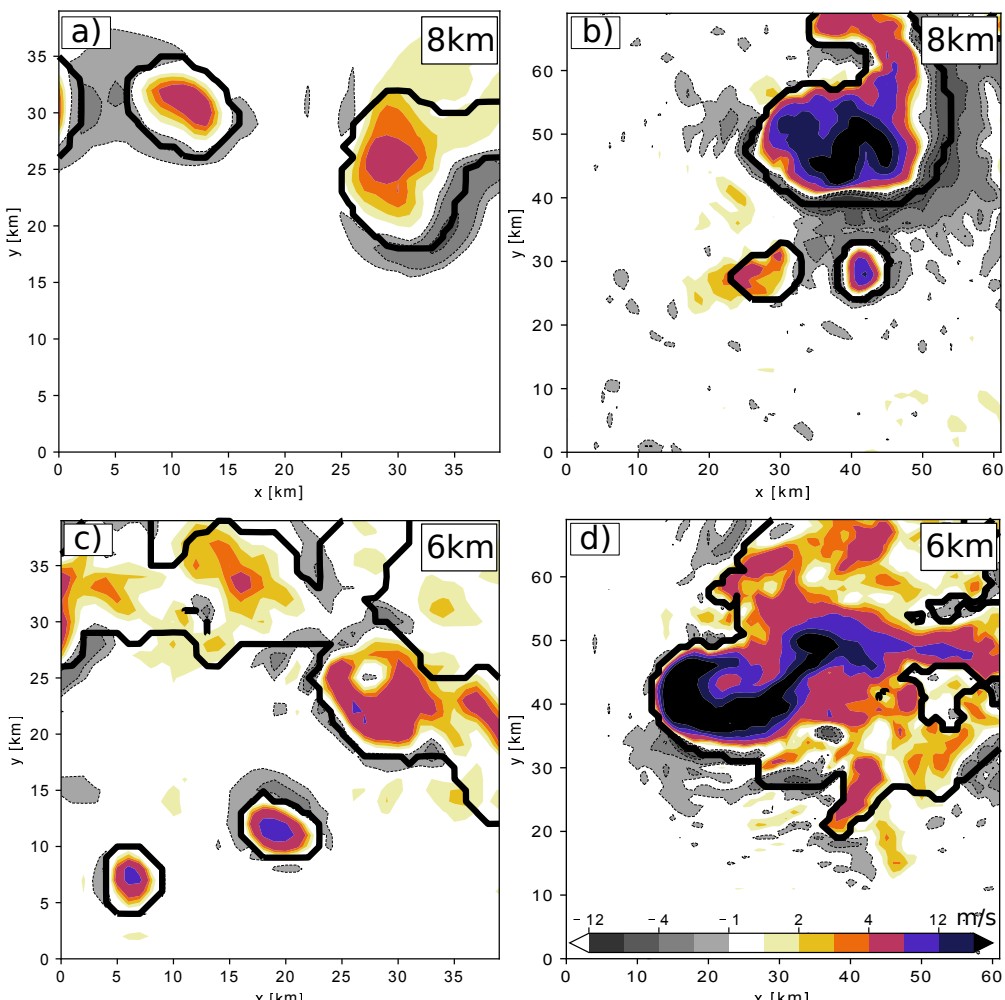

**Figure 12.** Horizontal cross section of vertical velocities at 6 km and 8 km height throughout the convective cells displayed in Figure 11 for : **(a)** and **(c) LR450**, and **(b)** and **(d) HR150**, respectively. The black solid line delineates the cloud boundaries (threshold of cloud and ice water contents $> 0.001$ g kg$^{-1}$).

At 150 m horizontal resolution, these eddies, as well as the updraft cores, are becoming better resolved as ascents exceed 12 m s$^{-1}$ over large areas. Furthermore, the strongest updraughts are neighboured by strong downdraughts (exceeding 10 m s$^{-1}$) just outside the cloud-edge that might be associated with a subsiding shell (Figure 12b and 12d). At 150 m the unresolved flow is mainly located at cloud edges and a significant part of the TKE contribution comes from the 3D dynamical production linked to the entrainment process. As a matter of fact, a clearly signature is simulated along the cloud-edge in HR150 (Figure 14d). As a consequence, it is possible to argue that the entrainment process, especially along the cloud-edge, is strongly underestimated

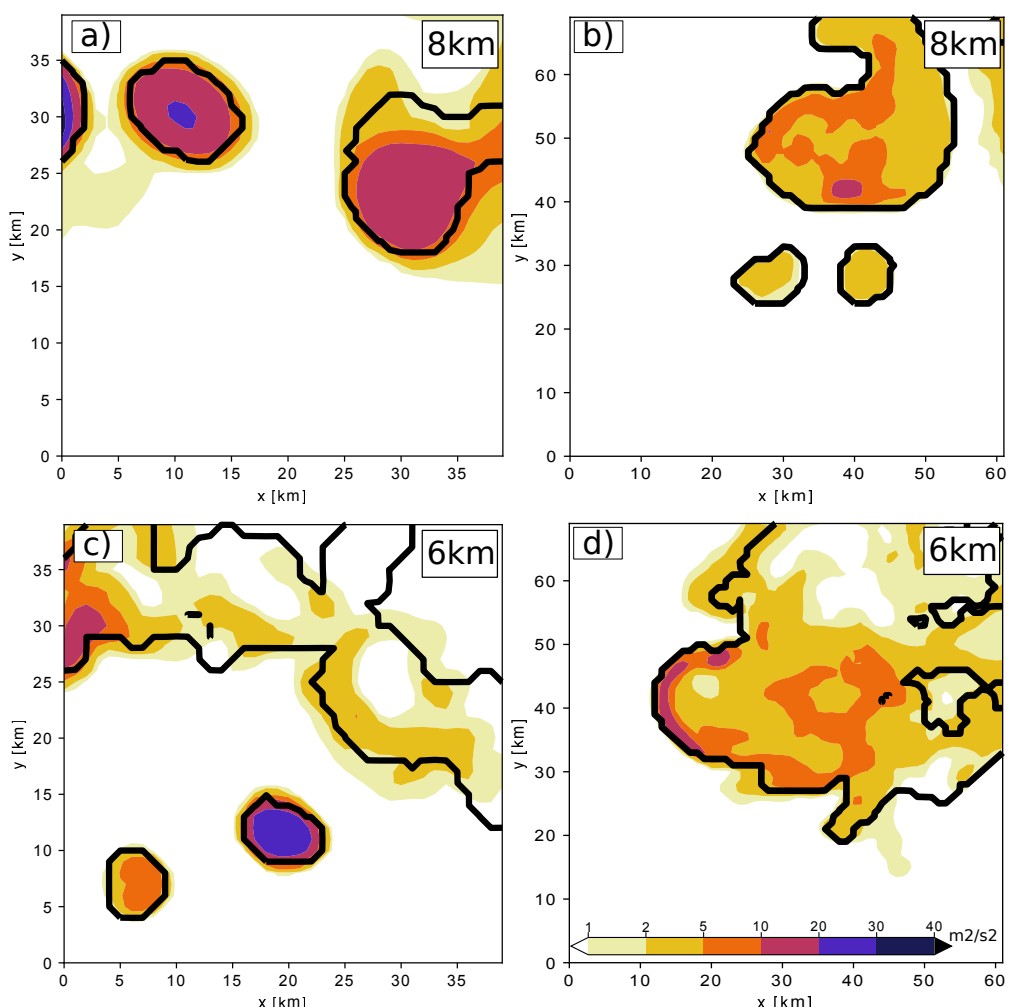

**Figure 13.** Same as Figure 12 but for the subgrid turbulent kinetic energy (TKE).

at 450 m horizontal resolution, that might lead to less entrainment of dryer environmental air in the clouds and thus LR450 simulates a too rapid development of the convective system and greater surface rainfall compared to HR150.

This issue of representation of entrainment between clouds and their environment at LES scale has been also assessed by other previous studies (Bryan et al., 2003; Heus et al., 2009; Khairoutdinov et al., 2009; Glenn and Krueger, 2014, among others). For example, Heath et al. (2017) also compared LES with horizontal resolutions of 150 m and 450 m for a continental deep convection case. They found that, even if moving from a kilometric grid spacing to LES does improve the representation of their case study, their LES-150 m does not adequately represent the associated small-scale forcing mechanisms. Moreover, in that case, higher resolutions worsen results due to relatively more environmental entrainment and moisture updraughts appear thus too diluted. In the other hand some previous studies also confirm the point that LES are primarily impacted

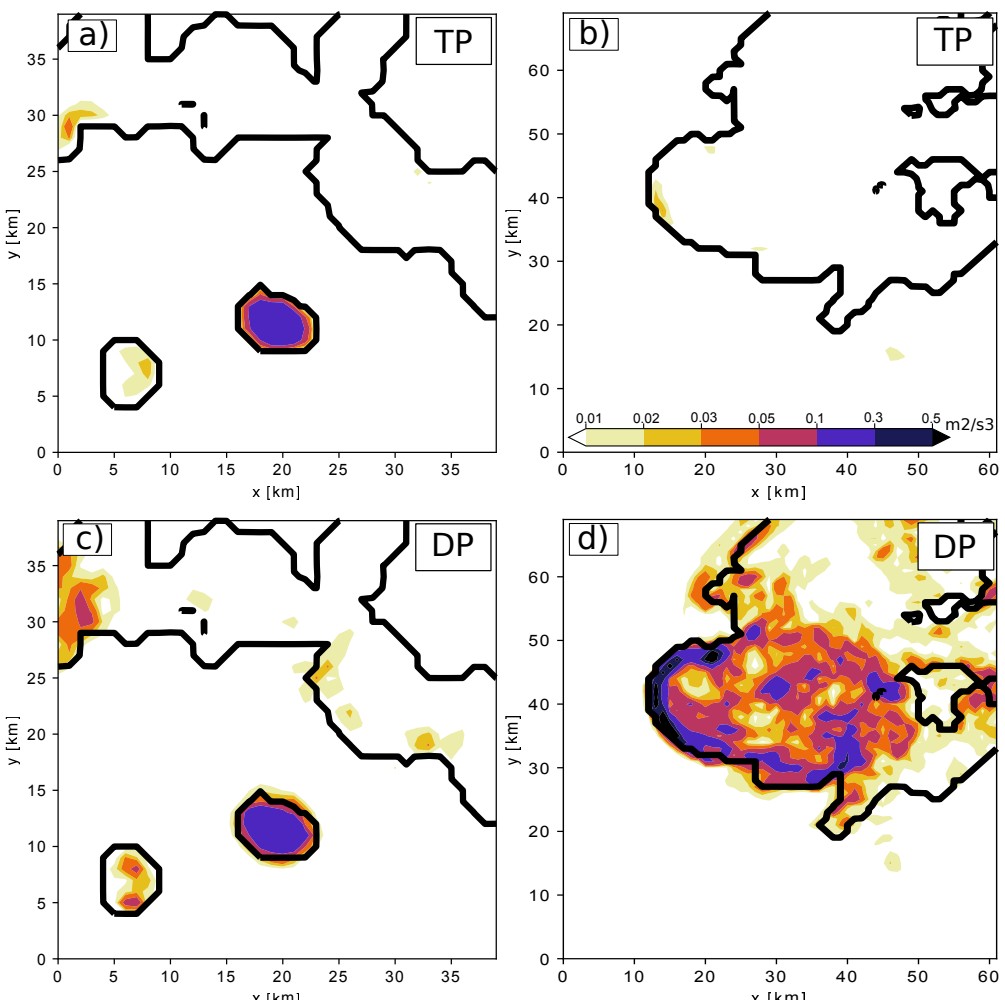

**Figure 14.** Horizontal cross section of thermal production (TP) and dynamical production (DP) at 6 km height throughout the convective cells displayed in Figure 11 for : **(a)** and **(c) LR450**, and **(b)** and **(d) HR150**, respectively.

by the mesoscale meteorological forcing through the lateral boundary conditions. As a result, even a LES with a horizontal resolution of 50 m does not significantly improve the quantitative precipitation forecast (Talbot et al., 2012). These previous studies confirm that increase horizontal resolution to LES grid spacing is necessary to better represent the small-scale processes governing deep convection organization, even if there is still further progress to be made in current parameterizations at this scale.

Furthermore, the LES simulation also better represent the spatial and temporal convective organization at cell scale within the precipitating system. Figure 15 illustrates a 3D rendering of the convective cell simulated by **HR150** between 0900 UTC and 0940 UTC. One can see at 0900 UTC a well developed cumulus cloud, and forming a well simulated arcus cloud just

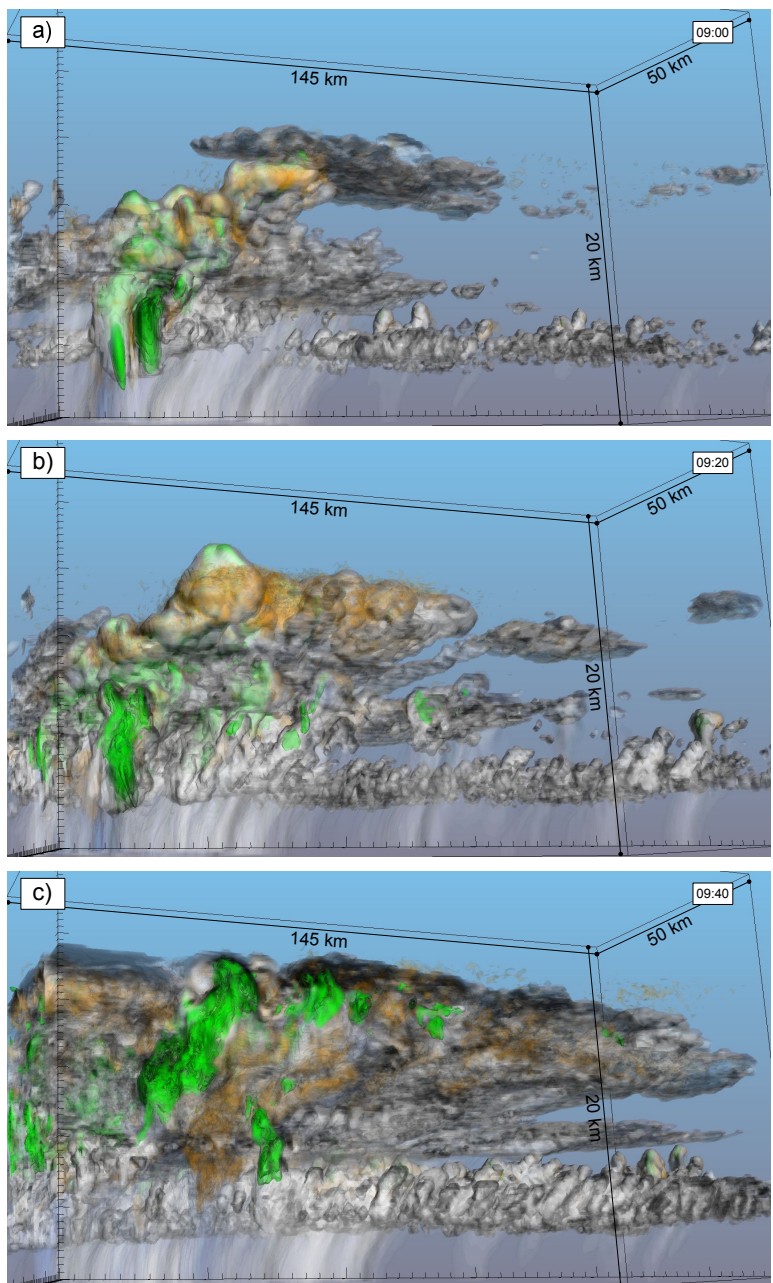

**Figure 15.** 3D rendering of the simulated convective cell from **HR150** between 0900 and 1000 UTC, in terms of rain water, cloud water, graupel, snow aggregate and ice water contents, respectively. Panels also show updraughts exceeding 5 m s$^{-1}$ (green colour) and downdraughts stronger than 2 m s$^{-1}$ (orange colour), respectively.

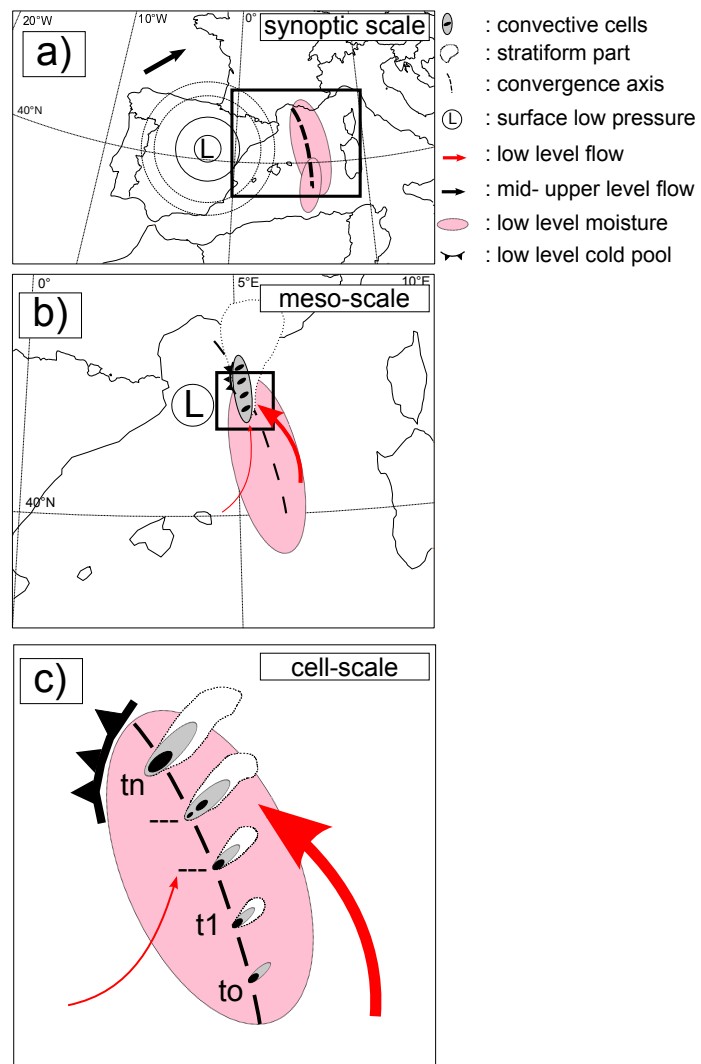

**Figure 16.** Schematic drawings of the precipitation structures of the convective systems and triggering ingredients observed during HyMeX IOP16a on 26 October 2012, at : **(a)** synoptic scale, **(b)** mesoscale, and **(c)** cell scale, respectively.

in front of very intense rainfall underneath the storms (Figure 15a). It is possible to track the cell across the following ten minutes. It continues to expand spatially and vertically in the following ten minutes while developing a stratiform part and stronger subsidence downstream (Figure 15b). Finally around 0940 UTC, the convective cell reaches its mature stage (Figure 15c), before loosing its identity and it is no longer discernible as it merges gradually with the rest of the precipitating system

5   (not shown) .

An animation of this 3D rendering performed with very high temporal resolution (every one minute) is very useful to better understand the 3D circulation within the precipitating system. The animation is available at https://doi.org/10.6096/mistrals-hymex.1540

(Nuissier, 2019). The southwest to southeasterly low level flow initiates convective cells just at the inner edge of the resulting strong convergence that, hereafter, propagate northwards while developing in the southwesterly upper level flow.

## 6   Conclusions

This study examines the impact of increasing horizontal resolution to Large Eddy Simulation (LES) in a numerical simulation of a real case study of Mediterranean HPE. For that purpose and for the first time, a large domain encompassing the convective systems, as well as the low level flow feeding convection over the Mediterranean sea, is considered. The goal here was to assess precisely how the physical mechanisms and convective organization are represented from sub-kilometric scale down to LES horizontal resolution. The paper focuses on an offshore convection case study, which took place on 26 October 2012 during the HyMeX SOP1. Figure 16 summarises the precipitation structures of the convective systems as well as the triggering mechanisms analysed during this HyMeX case study.

The convective systems observed during IOP16a were fed all along their evolution over the sea by moist and conditionally unstable air masses. A southwest to southeasterly converging low level flow is the main triggering mechanism acting to continually initiate and maintain the renewal of convective cells, contributing to a back-building-shaped system (Figure 16b). The low level convergence was enhanced strongly by a surface low pressure located between Spain and Balearic Islands (Figure 16a).

First, a LES carried out at 150 m horizontal resolution is compared, at the same scale, to another simulation performed with a 450 m grid spacing. On one hand, the increase of horizontal resolution from 450 m until 150 m is not able to improve significantly, for this case study, deficiencies of the simulation. Indeed, the simulated converging low level flow is quite similar in both simulations and only a single convective system is represented by both simulations instead of two compared to observations.

Although the present study does not present any sensitivity experiments to assess their precise role, initial and lateral boundary conditions might impact the simulations. The predictability of this heavy precipitation event, associated with offshore deep convection over the sea, is relatively low compared with more classical events anchored over the mountain range foothills. The direct orographic forcing appears less crucial while the convective systems were moving over the sea, but the neighbouring mountains are able to deflect the environmental mesoscale flow. Moreover, the model physics could also have a strong impact on the simulations. As a matter of fact, Martinet et al. (2017) showed for this case study that the formulation of the mixing length impacts the simulated surface precipitation through, in some cases, greater low levels moisture advection and hydrometeor contents within the convective system. Moreover, Thévenot et al. (2016) and Rainaud et al. (2017) even showed that taking into account the wave effect or sea surface conditions in different parameterizations of the sea state is able to modify locally the spatial distribution of the precipitation, although the overall rainfall pattern is globally well reproduced.

All these aspects are important but it must be emphasized that, during IOP16a case, the location and the evolution of deep convection over the sea (in particular the split into two distinct systems CS1 and CS2) are closely controlled by the upstream conditions (i.e. low levels moisture convergence generated by a surface low pressure located between Spain and Balearic Islands) and how they propagate inside of the LES domains. This split of deep convection over the sea is a real challenge

for this case study. Another numerical experiment could consider a larger LES domain encompassing these upstream conditions. Although this LES over a very large domain would suffer from expansive computing time, it would be able to address whether a higher resolution simulation of these features is crucial. Furthermore, there were numerous dedicated observations, in particular over the Mediterranean sea, during HyMeX SOP1 that captured fairly well the wind and moisture spatial and

vertical features of the upstream flow heading towards the French Mediterranean coastal regions (Duffourg et al., 2016). A posteriori assimilation of these field research observations could improve the quality of kilometric scale analyses arising from AROME-WMED for example. As a matter of fact, a reanalysis, including HyMeX observations, was carried out recently with AROME-WMED (Fourrié et al., 2019). New initial and lateral boundary conditions providing by this reanalysis may help improve the representation of the mesoscale flow over the sea for this case study.

On the other hand, the increase of horizontal resolution modifies the representation of some of triggering and organizing mechanisms controlling the precipitating system. More intense low level cold pools are simulated with a horizontal resolution of 450 m, probably related to evaporation of greater falling precipitation at least at the beginning of the simulation. As a matter of fact, at 150 m the unresolved flow is mainly located at cloud edges and a significant part of the turbulent kinetic energy contribution comes from the 3D dynamical production linked to the entrainment process. As a consequence, it is possible to

argue that the entrainment process, especially along the cloud-edge, is strongly underestimated at 450 m horizontal resolution, that might lead to less entrainment of dryer environmental air in the clouds that could explain why LR450 simulates a too rapid development of the convective system, greater surface rainfall and stronger low level cold pools compared to HR150.

Obviously, the results presented here need to be confirmed considering more convective case studies and using more statistical approaches. However, this first LES of a real Mediterranean precipitating case study highlighted an organization in

fast-propagating and developing cell trains within the converging low level flow (Figure 16), features that are definitively out of range of the kilometric resolution. In a general way, the goal of ongoing and future works is to better represent in the models at hectometric scales the key processes, such as turbulence and microphysics, that are crucial to progress in heavy precipitation forecasts.

*Acknowledgements.*  This work represents a contribution to the HyMeX program supported by MISTRALS and ANR MUSIC Grant ANR-
14-CE01-0014. The authors are grateful to all the scientists involved in the HYMEX SOP1 field campaign. Finally, we would like to thank Dr. Tim Dunkerton, as well as the two reviewers, for their precise and constructive remarks, which significantly helped improve the manuscript.

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
