# Peer review of "Hectometric scale simulations of a Mediterranean heavy precipitation event during HyMeX SOP1"

_Atmospheric Chemistry and Physics, 2019_

## Referee Comment (RC1) · Anonymous Referee #1 · 6 Dec 2019

General comments

This paper compares simulations of an offshore Mediterranean mesoscale convective system having different resolutions and analyses the triggering and organizing factors. It was found that increasing the horizontal resolution did not significantly reduce the deficiencies found in the coarser grid simulation. The reason is stated to be due primarily to issues with initial and lateral boundary conditions. It was noted that more realistic fine scale features were obtained with the higher resolution simulation. Overall this is a well organized paper with one of the main goals being to determine if a significantly higher resolution improves the models agreement with observations. It achieves that

goal even though the result is that very little improvement is gained for this particular case. Additionally a conceptual model is presented of the main physical processes taking place as the system evolves. My main concern is that it is stated that initial and lateral boundary conditions are the culprit, without any evidence provided that this is indeed the problem. It would help if the problems that are thought to exist with the initial and lateral boundary conditions and how they might be detrimentally affecting the simulation were discussed more thoroughly. Moreover, it seems likely that deficiencies with the model physics could negatively impact the simulation, and this should also be discussed. Is it possible that topography in this region, which is typically not handled very well by models if it is steep, is degrading the simulation? Finally there are numerous grammatical errors and unusual phrasing throughout the text that need to be addressed, as well as some issues with the figures.

Specific comments

1) Page 2, line 22: This is a very awkward sentence. Maybe change to something like "While these previous studies have shown that a horizontal resolution of about one kilometer is able to simulate many of the observed features of Mediterranean HPEs, as well as their associated key physical mechanisms, they have difficulties representing the time at which convection is triggered and its organization."

2) Page 2, line 27: "carried out numerical simulation of HPE case study". Might sound better to say "carried out numerical modeling HPE case studies".

3) Page 2, line 28: "similarly with". Might sound better to say "similar to".

4) Page 2, line 34: "kilometers" should me "meters".

5) Page 3, line 1: "depending of' should be "depending on".

6) Page 3, line 7: "with eddy-diffusivity" should be "with the eddy-diffusivity".

7) Page 3, line 14: "calls to". Unusual wording. Maybe say "motivates the adoption of a Large . . ." 8) Page 3, line 21: change "already" to "previously".

9) Page 3, line 23: maybe change "truthful" to "realistic". Although your conclusion is that there may be issues with the initial and boundary conditions, so they are unlikely to be as realistic as they need to be.

10) Page 4, line 15: maybe change "to some" to "for".

11) Page 6, line 3: change "an unique" to "a unique".

12) Page 6, line 10: "mentionned". Spelling is incorrect here and elsewhere.

13) Page 7, line 4: "The vertical grid is streched with a spacing not varying continuously with altitude". Streched should be spelt stretched, and maybe change the sentence to "The vertical grid spacing is stretched with altitude".

14) Page 7, line 13: Change "while other 20" to "while the other".

15) Page 7. The model description should include mention of the radiation, surface flux, and topography schemes, even if it is just to say that they are the same as used in a previous study.

16) Page 7. Line 30: change "An other" to "Another" here and elsewhere in the text.

17) Page 8, line 8: "over sea" should be "over the sea". And elsewhere.

18) Page 8 line 9: "It allows to better understand and model the triggering mechanisms within the convective systems." Awkward wording.

19) Page 8, line 15: "fairly simulate". Maybe "simulate fairly well".

20) Page 8, line 4: "These large surface precipitation are induced by the convective systems CS1 and CS2 mentionned earlier". Maybe change to "These regions of large precipitation are caused by . . .".

21) Page 9, line 12: "This domain enables to encompass the evolution of the convective system over sea, not considering here mainland precipitation." Spelling mistake and awkward wording.

22) Page 9, line 14: "exceeding" should be "exceeds".

23) Page 10, line 2: "examinated" "examined".

24) Page 10, line 11: "for a too long time" "for too long".

25) Page 10, line 11: "essentially due to the lateral boundary conditions imposed by AROME-WMED analysis". How do you know this?

26) Page 11, line 1: "just east tip of Spain". Needs to be reworded.

27) Page 11, line 11: "upper level" "upper levels".

28) Page 11, line 13, "organizing in" "organizing into".

29) Page 11, line 23: "development evolution". Choose one of these words.

30) Page 11, line 29: "of Balearic Islands" "of the"

31) Page 11, line 31: "already" maybe "previously" . 32) Page 11, line 32: "than those" "to those".

33) Page 12, line 3: "updraught" "updraughts".

34) Page 12, line 9: "updraught" "updraughts".

35) Page 13, line 1: "favouring lifting more easily". Awkward wording.

36) Page 13, line 2: "enhance" "enhances".

37) Page 16, line 1: "resolution than" "resolution as".

38) Page 16, line 2: "in depth" maybe "significantly".

39) Page 16, line 4: "rainfall evaporation under the convective system" maybe change to something like "rainfall evaporation at low-levels in the convective system", since the rain is part of the convective system and not under it.

40) Page 16, line 5: "more precisely dynamics" "the dynamics more precisely".

41) Page 16, line 11" "thanks to" "by".

42) Figure 10: It is hard to make out how the radar relates to the model results. It looks like they have different scales.

43) Page 16, line 26: "abutting"?

44) Page 16, line 33: "in a too abruptly manner". Need to reword.

45) Page 17, line 3. "enables to emphasize"> Need to reword.

46) Figure 12. Is there any way this can be improved. What is the horizontal and vertical scale?

47) Page 16, line 2: "just front of very". Need to reword.

48) Page 20, line 18: "over Mediterranean" "over the".

49) Page 20, line 18: "captured fairly" "captured fairly well".

50) Page 20, line 29: "could be is". Need to reword.

---

## Referee Comment (RC2) · Anonymous Referee #2 · 9 Dec 2019

General comments

This manuscript presents a 150 m horizontal grid spacing 12h LES simulation over a relatively large area on the Mediterranean basin of a deep moist convection real case (HyMeX SOP1 IOP16a) (offshore MCS moving towards SE France). It is a valuable contribution to the field because it describes, as stated by the authors, the first LES of a heavy rainfall event over the Mediterranean area, with the added value of selecting an event with a complete observational data set collected during the HyMeX SOP1 field campaign to verify the simulation. The structure of the manuscript is adequate and the text is generally clear despite in some parts some minor corrections are necessary -

see specific details below. I also think that the discussion of the results would benefit from considering the following two aspects.

Firstly, authors could compare in more detail the simulations with available observations (for example weather radar or satellite data): some parts of the text seem more focused on comparing the 150 m simulation with the 450 m simulation than with the observations available, probably authors performed this analysis but it could be better reflected in the text and in figures (for example figures 5, 6 and 7 would largely benefit from showing also observations as it is done in figure 4).

Secondly, authors provide a good comparison of their results with previous simulations of the same case study, which is very interesting, but they could also, at least briefly, compare their results with other similar previous studies (either ideal or real cases) using LES - I include a few in the references but others could be considered. For example Heath et al also compare 150 m with 450 m LES of a deep convection case and find that higher resolutions worsen results because entrainment is more important and apparently moisture updrafts are too diluted.

For all the above I believe this manuscript must be considered for publication but cannot be accepted in its current form. I encourage authors to consider these general and also specific comments below for preparing and submitting an improved version of this very interesting paper.

Specific comments

1. Page 2, line 14. Suggest: the deep convection -> deep convection

2. Page 3, line 6 (and elsewhere). You write "500m" here without a blank space between the number and the units but elsewhere you leave a blank space (i.e. page 1, line 5) - please be consistent.

3. Page 3, line 19. Please add the year of the second reference cited in this line.

4. Page 3, line 20. Do you mean "the authors's" (plural) or really a specific author

(singular)?

5. Page 3, line 21. Please check "has already been performed"-> "has been performed yet"

6. Page 4, figure caption 1. Please specify which infrared channel was used.

7. Page 6, line 5. typo: last word should be 'conditionally'

8. Page 6, line 10 (and elsewhere). Typo: mentionned -> mentioned

9. Page 7, line 5. Please check sentence (2 items): 1st item) streched -> stretched? and 2nd item) "with a spacing not varying continuously with altitude" do you mean that or simply "with a irregular spacing"? Please clarify and check meaning.

10. Page 7, line 30. was retained -> were used? Please check meaning.

11. Page 7, line 30 (and elsewhere): an other -> another

12. Page 8, line 1 (and elsewhere): untill -> until

13. Page 8, line 15. Please check meaning: fairly simulate -> simulate fairly well

14. Page 9, figure 5b. I find a bit misleading the y-axis title, and also the fact that it is not the same as that used in the figure caption; I suggest using "Surface precipitation coverage" or simply "Precipitation coverage" instead. Moreover, as mentioned earlier I recommend to add, to both panels, the corresponding observed magnitude derived from weather radar observations.

15. Page 9, line 14. suggest: an area -> the area; and also exceeding -> exceeded

16. Page 10, figure 6. Please add here another column with the observed radar reflectivities (blank spacing between the 3 columns may be removed to allow larger panels)

17. Page 11, line 8. et -> and

18. Page 11, line 10. Typo: whith -> with

19. Page 12, figure 7. Please add another column with the corresponding satellite observations.

20. Page 12, line 3 (and elsewhere). suggest larger -> greater

21. Page 12, line 9. Typo: updraught -> updraughts

22. Page 13, line 10. Suggest: 50% more the values for LR150 -> 50% the LR150 values

23. Page 15, figure 10. The comparison of the four panels is difficult without a common geographical grid or a length scale: in panel a) the grid is missing, in panels b) and c) there is one grid despite it is not the same, and in panel d) it is not obvious which part should be compared. Please add either a grid (preferably) or a length scale to each panel.

24. Page 17, figure 11 caption first line. Check: crossection -> cross section

25. Page 18, figure 12. Is precipitation also shown in the figure panels? If so please indicate it on the figure caption.

26. Page 20, line 18. fairly -> fairly well?

27. Page 20, line 30. be is -> be

28. Page 20, line 32. "combining and mixing"? I do not understand what authors mean by 'mixing' here, please clarify.

29. Page 24, please check and correct reference Martinet et al: author names duplicated and title missing.

References

Heath, N. K., Fuelberg, H. E., Tanelli, S., Turk, F. J., Lawson, R. P., Woods, S., & Freeman, S. (2017). WRF nested large‐eddy simulations of deep convection during

SEAC4RS. Journal of Geophysical Research: Atmospheres, 122(7), 3953-3974.

Khairoutdinov, M. F., Krueger, S. K., Moeng, C. H., Bogenschutz, P. A., & Randall, D. A. (2009). Large‐Ředdy simulation of maritime deep tropical convection. Journal of Advances in Modeling Earth Systems, 1(4).

Talbot, C., Bou-Zeid, E., & Smith, J. (2012). Nested mesoscale large-eddy simulations with WRF: performance in real test cases. Journal of Hydrometeorology, 13(5), 1421-1441.

---

## Author Comment (AC1) · 13 Feb 2020

General comments:

1) More comparison against observations:

We agree that a deeper analysis comparing the simulations and the available observations should improve the paper. However, it must be emphasized that such analysis has been performed previously in the studies of Duffourg et al, 2016 or Martinet et al, 2017, for which the numerical design was similar. Although the goal of the present study is to focus more on comparing the 150 m simulation with the 450 m simulation,

additional observations have been added in the revised Figure 6 and 7, as it is done in Figure 4. On the other hand we did not modify former Figure 5 since it is more difficult to compare the simulations to a radar quantitative surface precipitation estimate strongly impacted by large uncertainties over the sea.

2) Compare the results with other similar studies:

We agree that compare the results with other similar studies using LES should improve the paper. Furthermore we are grateful to the referee #2 for providing a list of references which has been included in the revised version of the paper.

Specific comments:

1) Page 2, line 14: The text has been corrected.

2) Page 2, line 27: This problem has been corrected throughout the text in order to be consistent.

3) Page 3, line 19: The year of the reference has been added.

4) Page 3, line 20: We meant plural. The text has been modified accordingly.

5) Page 3, line 21: The text has been corrected.

6) Page 4, figure caption 1: The IR channel used here is 9 (10.8 $\mu$m). The figure caption has been modified accordingly.

7) Page 6, line 5: The text has been corrected.

8) Page 6, line 10: The text has been corrected.

9) Page 7, line 5: The sentence has been rewritten.

10) Page 7, line 30: The text has been corrected.

11) Page 7, line 30 (and elsewhere): The word has been corrected here and throughout the text.

12) Page 8, line 1 (and elsewhere): The word has been corrected here and throughout the text.

13) Page 8, line 15: The text has been corrected.

14) Page 9, Figure 5b: The y-axis title has been modified accordingly. Regarding the remark of including a surface precipitation estimation from radar observations see please response to general comments above.

15) Page 9, line 14: The text has been corrected.

16) Page 10, Figure 6: Figure 6 has been redrawn adding another column with the observed radar reflectivities.

17) Page 11, line 8: The text has been corrected.

18) Page 11, line 10: The text has been corrected.

19) Page 12, Figure 7: Figure 7 has been redrawn adding another column with the observed infrared satellite observations.

20) Page 12, line 3 (and elsewhere): The word has been corrected here and throughout the text.

21) Page 12, line 9: The text has been corrected.

22) Page 13, line 10: The text has been corrected.

23) Page 15, Figure 10: We agree with this remark. It is not obvious to compare all simulations against observations as the simulation domains are different initially. However former Figure 10 has been redrawn zooming over the region of interest for the observations and including a grid on each panel.

24) Page 17, Figure 11: The caption has been corrected.

25) Page 18, Figure 12: All the hydrometeor contents are represented (rain water, cloud water, graupel, snow aggregate and ice water contents). The caption has been

modified accordingly.

26) Page 20, line 18: The text has been corrected.

27) Page 20, line 30: The text has been corrected.

28) Page 20, line 32: We meant that we need to consider more convective case studies and use more statistical approaches. The sentence has been rewritten.

29) Page 24: The reference has been corrected.
* * *

---

## Author Comment (AC2) · 17 Feb 2020

General comments:

We fully agree with this general remark. In this present study, we recognize that we are not showing any sensitivity experiments to assess the precise role played by the lateral boundary conditions and we also recognize that the conclusion presented here is too affirmative. The predictability of this heavy precipitation event, associated with offshore deep convection over the sea, is relatively low compared with more classical events anchored over the mountain range foothills. The direct orographic forcing appears less crucial while the convective systems were moving over the sea, but the neighbouring mountains are able to deflect the environmental mesoscale flow. We agree that the model physics could also have a strong impact on the simulation. As a matter of fact, Martinet et al. (2017) showed for this case study that the formulation of the mixing length impacts the simulated surface precipitation through, in some cases, greater low levels moisture advection and hydrometeor contents within the convective system. Moreover, Thévenot et al. (2016) and Rainaud et al. (2017) even showed that taking into account the wave effect or sea surface conditions in different parameterizations of the sea state are able to modify locally the spatial distribution of the precipitation, although the overall rainfall pattern is globally well reproduced.

We agree that all these aspects are important but it must be emphasized that, during IOP16a case, the location and the evolution of deep convection over the sea (in particular the split into two distinct systems CS1 and CS2) are closely controlled by the upstream conditions (i.e. low levels moisture convergence generated by a surface low pressure located between Spain and Balearic Islands) and how they propagate inside of our LES domains. This split of deep convection over the sea is a real challenge for this case study. Another numerical experiment could consider a larger LES domain encompassing these upstream conditions. Although this LES over a very large domain would suffer from expansive computing time, it would be able to address whether a higher resolution simulation of these features is crucial. All these aspects have been included and discussed in the revised version of the paper.

Specific comments:

1) Page 2, line 22: The sentence has been rewritten.

2) Page 2, line 27: The text has been corrected.

3) Page 2, line 28: The text has been corrected.

4) Page 2, line 34: The text has been corrected.

5) Page 3, line 1: The text has been corrected.

6) Page 3, line 7: The text has been corrected.

7) Page 3, line 14: The text has been corrected.

8) Page 3, line 21: The text has been changed.

9) Page 3, line 23: The text has been changed.

10) Page 4, line 15: The text has been changed.

11) Page 6, line 3: The text has been changed.

12) Page 6, line 10: The word has been corrected here and throughout the text.

13) Page 7, line 4: The sentence has been rewritten accordingly.

14) Page 7, line 13: The text has been changed.

15) Page 7. Additional parametrization schemes have been added in the model description accordingly.

16) Page 7, line 30: The word has been corrected here and throughout the text.

17) Page 8, line 8: The word has been corrected here and throughout the text.

18) Page 8, line 9: The sentence has been rewritten.

19) Page 8, line 15: The text has been corrected.

20) Page 8, line 4: The text has been changed.

21) Page 9, line 12: The sentence has been rewritten.

22) Page 9, line 14: The text has been corrected.

23) Page 10, line 2: The text has been corrected.

24) Page 10, line 11: The text has been corrected.

25) Page 10, line 11: We agree with this remark. We cannot state "essentially due to

the lateral boundary conditions...". The sentence has been rewritten (see also response to general remark).

26) Page 11, line 1: The sentence has been rewritten.

27) Page 11, line 11: The text has been corrected.

28) Page 11, line 13: The text has been corrected.

29) Page 11, line 23: The text has been corrected.

30) Page 11, line 29: The text has been corrected.

31) Page 11, line 31: The text has been corrected.

32) Page 11, line 32: The text has been corrected.

33) Page 12; line 3: The text has been corrected.

34) Page 12, line 9: The text has been corrected.

35) Page 13, line 1: The sentence has been rewritten.

36) Page 13, line 2: The text has been corrected.

37) Page 16, line 1: The text has been corrected.

38) Page 16, line 2: The text has been corrected.

39) Page 16, line 4: The sentence has been rewritten.

40) Page 16, line 5: The text has been corrected.

41) Page 16, line 11: The text has been corrected.

42) Figure 10: We agree with this remark. It is not obvious to compare all simulations against observations as the simulation domains are different initially. However former Figure 10 has been redrawn zooming over the region of interest on the observations and including a grid on each panel.
43) Page 16, line 26: The text has been corrected.

44) Page 16, line 33: The sentence has been rewritten.

45) Page 17, line 3: The sentence has been rewritten.

46) Figure 12: In the revised version former Figure 12 has been improved adding the horizontal and vertical scale.

47) Page 8, line 1: The text has been corrected.

48) Page 20, line 18: The text has been corrected.

49) Page 20, line 18: The text has been corrected.

50) Page 20, line 29: The text has been corrected.

---

## Author Response (AR2)

**REPLY TO THE COMMENTS OF EDITOR**

**General comment: lack of clarification and illustration to better understand how the physical involved mechanisms are modified/improved while increasing horizontal resolution from 450 m to 150 m.**

As a matter of fact the main purpose of the paper is effectively to assess how the physical mechanisms and convective organization, involved in a real case study of Mediterranean heavy precipitation, are represented from 450 m down to 150 m. The results obtained in this present study confirm the need to use horizontal grid spacing of about a hundred of meters to resolve properly (at least partly explicitly) these eddies containing most of the kinetic energy and thus to better represent in particular the different stages of convection, especially the possible link with cloud-edge entrainment.

One of the main conclusions of the paper is that increase of horizontal resolution does not significantly impact the larger scale forcing in which deep convection is embedded (especially the southwest- to southeasterly low-level moist and converging flow driven by a surface low pressure between Spain and Balearic Islands), and the location of the convective systems themselves (more related to initial and lateral boundary condition issues). However, when focussing at convective cell scale, HR150 appears more realistic in terms of cloud appearance and evolution, whereas deep convection in LR450 could be triggered too abruptly, leading to enhanced microphysical processes and amplified mechanisms (stronger low level cold pools). We fully agree that there was a lack of illustration regarding that point and we are grateful to the editor for his help providing a list of possible fields to be drawn.

For that purpose, a few additional figures (cf. new Figure 12, 13 and 14) have been added in the revised version of the paper, showing horizontal cross sections of vertical velocities, subgrid turbulent kinetic energy (TKE) at 6 km and 8 km height, and both dynamical and thermal contributions in the TKE production at 6 km height, throughout the convective cells displayed in Figure 11, respectively. As one can see in Figure 12a,12c and Figure 13a,13c at 450 m horizontal resolution, the updraft cores are partly resolved, as vertical velocities do not exceed 12 m s$^{-1}$, and are also partly unresolved, with TKE sometimes higher than 30 m² s$^{-2}$. This TKE is produced by both dynamical and thermal processes, and is mainly localized in updraft cores. The eddies near the cloud edges, which are subgrid at 450m resolution, are not represented by the turbulence scheme. These results are in a good agreement with those obtained by Verrelle et al. (2017) and Strauss et al. (2019), who have shown that a commonly used eddy-diffusivity turbulence scheme underestimates the TKE at kilometric and hectometric (500m) resolution, especially at the cloud edges but also in the updraft

cores where the thermal production is misrepresented because the scheme does not enable the countergradient structures present in the updraft to be reproduced. At 150 m resolution, these eddies, as well as the updraft cores, are becoming better resolved as ascents exceed 12 m s$^{-1}$ over large areas (Figure 12b,12d) . Furthermore, the strongest updraughts are neighboured by strong downdraughts (exceeding 10 m s$^{-1}$) just outside the cloud-edge that might be associated with a subsiding shell (Figure 12b,12d). At 150 m the unresolved flow is mainly located at cloud edges and a significant part of the TKE contribution comes from the 3D dynamical production linked to the entrainment process, with a clear signature visible in Figure 14d . As a consequence,  it is possible to argue that the entrainment process, especially along the cloud-edge, is strongly underestimated at 450 m horizontal resolution, that might lead to less entrainment of dryer environmental air in the clouds that could explain why LR450 simulates a too rapid development of the convective system, greater surface rainfall and stronger low level cold pools compared to HR150.

Regarding the cold-pool aspect, Figure 8 already illustrates differences in terms of horizontal low-level flow (moisture advection and low-level flow between 0 and 3 km height) between both 450 m and 150 m horizontal resolution around the cloud complexes. The simulated environmental flow is not significantly different between both simulations. However, we do think that comparison of horizontal motions within the convective systems appear less obvious and more complicated, as clouds are not simulated at the same location and the same time in both simulations that strongly disrupt the flow around.  Nevertheless, although other factors could impact the cold-cold, greater surface precipitation simulated at 450 m horizontal resolution lead to more cooling underneath the storms.

References :
- Verrelle, A., D. Ricard, and C. Lac, Evaluation and improvement of turbulence parameterization inside deep convective clouds at kilometer-scale resolution, *Mon. Weather Rev.*, *145*, 3947-3967, 2017.
- Strauss, C., D. Ricard, C. Lac, and A. Verrelle, Evaluation of turbulence parameterizations in convective clouds and their environment based on a large-eddy simulation, *Quart. J. Roy. Meteor. Soc.*, *145*, 3195-3217, 2019.